# Vertical Federated Learning with Missing Features During Training and Inference

**Pedro Valdeira**
Carnegie Mellon University
& Instituto Superior Técnico
pvaldeira@cmu.edu

**Shiqiang Wang**
IBM Research
wangshiq@us.ibm.com

**Yuejie Chi**
Carnegie Mellon University
yuejiechi@cmu.edu

## Abstract

Vertical federated learning trains models from feature-partitioned datasets across multiple clients, who collaborate without sharing their local data. Standard approaches assume that all feature partitions are available during both training and inference. Yet, in practice, this assumption rarely holds, as for many samples only a subset of the clients observe their partition. However, not utilizing incomplete samples during training harms generalization, and not supporting them during inference limits the utility of the model. Moreover, if any client leaves the federation after training, its partition becomes unavailable, rendering the learned model unusable. Missing feature blocks are therefore a key challenge limiting the applicability of vertical federated learning in real-world scenarios. To address this, we propose `LASER-VFL`, a vertical federated learning method for efficient training and inference of split neural network-based models that is capable of handling arbitrary sets of partitions. Our approach is simple yet effective, relying on the sharing of model parameters and on task-sampling to train a family of predictors. We show that `LASER-VFL` achieves a $\mathcal{O}(1/\sqrt{T})$ convergence rate for nonconvex objectives and, under the Polyak-Łojasiewicz inequality, it achieves linear convergence to a neighborhood of the optimum. Numerical experiments show improved performance of `LASER-VFL` over the baselines. Remarkably, this is the case even in the absence of missing features. For example, for CIFAR-100, we see an improvement in accuracy of $19.3\%$ when each of four feature blocks is observed with a probability of 0.5 and of $9.5\%$ when all features are observed. The code for this work is available at `https://github.com/Valdeira/LASER-VFL`.

## 1 Introduction

In federated learning (FL), a set of clients collaborates to jointly train a model using their local data without sharing it (Kairouz et al., 2021). In horizontal FL, data is distributed by samples, meaning each client holds a different set of samples but shares the same feature space. In contrast, vertical FL (VFL) involves data distributed by features, where each client holds different parts of the feature space for overlapping sets of samples. Whether an application is horizontal or vertical FL is dictated by how the data arises, as it is not possible to redistribute the data. This work focuses on VFL.

In VFL (Liu et al., 2024), the global dataset $\mathcal{D} \coloneqq \{\boldsymbol{x}^1, \ldots, \boldsymbol{x}^N\}$, where $N$ is the number of samples, is partitioned across clients $\mathcal{K} \coloneqq \{1, \ldots, K\}$. Each client $k \in \mathcal{K}$ typically holds a local dataset $\mathcal{D}_k \coloneqq \{\boldsymbol{x}_k^1, \ldots, \boldsymbol{x}_k^N\}$, where $\boldsymbol{x}_k^n$ is the block of features of sample $n$ observed by client $k$. We have that $\boldsymbol{x}^n = (\boldsymbol{x}_1^n, \ldots, \boldsymbol{x}_K^n)$. Unlike horizontal FL, in VFL, different clients collect local datasets with distinct types of information (features). Such setups—e.g., an online retail company and a social media platform holding different features on shared users—typically involve entities from different sectors, reducing competition and increasing the incentive to collaborate. To train VFL models without sharing the local datasets, split neural networks (Ceballos et al., 2020) are often considered.

In split neural networks, each client $k$ has a *representation model* $\boldsymbol{f}_k$, parameterized by $\boldsymbol{\theta}_{\boldsymbol{f}_k}$. The representations extracted by the clients are then used as input to a *fusion model* $g$, parameterized by $\boldsymbol{\theta}_g$. This fusion model can be at one of the clients or at a server. Thus, to learn the parameters $\boldsymbol{\theta}_\mathcal{K} \coloneqq (\boldsymbol{\theta}_{\boldsymbol{f}_1}, \ldots, \boldsymbol{\theta}_{\boldsymbol{f}_K}, \boldsymbol{\theta}_g)$ of the resulting *predictor* $h$, we can solve the following problem:

$$\min_{\boldsymbol{\theta}_\mathcal{K}} \quad \frac{1}{N} \sum_{n=1}^{N} \ell(h(\boldsymbol{x}^n; \boldsymbol{\theta}_\mathcal{K}), y^n) \quad \text{where} \quad h(\boldsymbol{x}^n; \boldsymbol{\theta}_\mathcal{K}) \coloneqq g\left(\{\boldsymbol{f}_k(\boldsymbol{x}_k^n; \boldsymbol{\theta}_{\boldsymbol{f}_k})\}_{k=1}^{K}; \boldsymbol{\theta}_g\right), \quad (1)$$

with $\ell$ denoting a loss function and $y^n$ the label of sample $n$, which we assume to be held by the same entity (client or server) as the fusion model. We illustrate this family of models in Figure 1a.

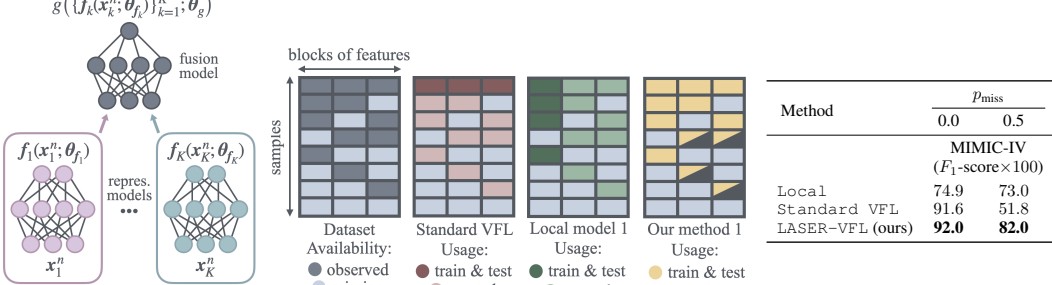

(a) Split neural network.  (b) Dataset availability and usage.  (c) Results preview.

Figure 1: In Figure 1a, we illustrate a split neural network. In Figure 1b, we show the availability of a dataset with $K = 3$ blocks of features, each with a $0.5$ probability of being observed, and its usage and waste by three key methods: standard VFL, a local approach, and our method. Standard VFL trains a single predictor, while the local approach and our method train different predictors at different clients (we show the data usage for client 1). In Figure 1c, we present a preview of our results: a mortality prediction task using the MIMIC-IV dataset, when the probability of each block of features missing, $p_{\text{miss}}$, is in $\{0.0, 0.5\}$, for both data and test data.

**Generalization and availability in standard VFL.**  We see in (1) that, even for a single sample $n$, predictor $h$ depends on all the blocks of features, $\{\boldsymbol{x}_k^n : \forall k\}$. Thus, for both training and inference, (1) requires the observations of all the clients to be available.[1] Building on our example of an online retailer and a social media company sharing users, each company is also likely to have unique users. This applies to both training and test data. Further, if any client drops from the federation during inference, its block will permanently stop being observed. In both the case of nonshared users and of clients leaving the federation, standard VFL predictors become unusable. Figure 1b illustrates this phenomenon: for both training and inference, if each of $K = 3$ clients observes its block of features with an (independent) probability of $0.5$, only $0.5^3 = 12.5\%$ of the original data is usable. Thus, restricting training to fully-observed samples hinders generalization, while restricting inference to such samples limits the availability and utility of the model.

**Dealing with missing features.**  To handle missing features in training data, some approaches expand the dataset, filling the missing features before collaborative training (Kang et al., 2022b), while others use their partition of partially-observed samples for local representation learning but exclude them from collaboration (He et al., 2024). These methods can outperform standard VFL when partially observed samples are present, but they add new training stages and auxiliary modules, making them more complex. Moreover, they train a joint predictor that requires the collaboration of all clients during inference. On the other hand, to improve robustness against missing blocks at inference, each client can train a local predictor using only its own features, avoiding collaboration altogether. This also addresses missing features in the training data. Alternatively, collaborative methods can employ techniques such as knowledge distillation (Huang et al., 2023) and data augmentation (Gao et al., 2024) to train local predictors, also adding complexity with multiple stages. However, while robust to missing features, these local predictors cannot utilize additional feature blocks if available at test time, resulting in wasted data and reduced predictive power. We illustrate this in Figure 1b.

Therefore, there is a gap in the literature when it comes to leveraging all the available data without either dropping incomplete samples or ignoring existing features. This raises the following question:

*Can we design an efficient VFL method that is robust to missing features at both training and inference with provable convergence guarantees, without wasting data?*

In this work, we answer this question in the affirmative. We propose LASER-VFL (**L**everaging **A**ll **S**amples **E**fficiently for **R**eal-world **VFL**), a novel method that enables both training and inference using any and all available blocks of features. To the best of our knowledge, this is the first method to achieve this. Our approach avoids the multi-stage pipelines that are common in this area, providing

---

[1]This contrasts with horizontal FL, where the local data of one client suffices to estimate mini-batch gradients and to perform inference.

a simple yet effective solution that leverages the sharing of model parameters alongside a task-sampling mechanism. By fully utilizing all data (see Figure 1b), our method leads to significant performance improvements, as demonstrated in Figure 1c. Remarkably, our approach even surpasses standard VFL when all samples are fully observed. We attribute this to a dropout-like regularization effect introduced by task sampling.

**Our contributions.** The main contributions of this work are as follows.

- We propose `LASER-VFL`, a simple, hyperparameter-free, and efficient VFL method that is flexible to varying sets of feature blocks during training and inference. To the best of our knowledge, this is the first method to achieve such flexibility without wasting either training or test data.
- We show that `LASER-VFL` converges at a $\mathcal{O}(1/\sqrt{T})$ rate for nonconvex objectives. Furthermore, under the Polyak-Łojasiewicz (PL) inequality, it achieves linear convergence to a neighborhood around the optimum.
- Numerical experiments show that `LASER-VFL` consistently outperforms baselines across multiple datasets and varying data availability patterns. It demonstrates superior robustness to missing features and, notably, when all features are available, it still outperforms even standard VFL.

**Related work.** VFL shares challenges with horizontal FL, such as communication efficiency (Liu et al., 2022; Valdeira et al., 2025) and privacy preservation (Yu et al., 2024), but also faces unique obstacles, such as missing feature blocks. During training, these unavailable partitions render the observed blocks of other clients unusable, and at test time, they can prevent inference altogether. Therefore, most VFL literature assumes that all features are available for both training and inference—an often unrealistic assumption that has hindered broader adoption of VFL (Liu et al., 2024).

To address the problem of missing features during VFL training, some works use nonoverlapping, or nonaligned, samples (that is, samples with missing feature blocks) to improve generalization. In particular, Feng (2022) and He et al. (2024) apply self-supervised learning to leverage nonaligned samples locally for better representation learning, while using overlapping samples for collaborative training of a joint predictor. Alternatively, Kang et al. (2022b), Yang et al. (2022), and Sun et al. (2023) employ semi-supervised learning to take advantage of nonaligned samples. Although these methods enable VFL to utilize data that conventional approaches would discard, they still train a joint predictor and thus require all features to be available for inference, which remains a limitation.

A recent line of research leverages information from the entire federation to train local predictors. This can be achieved via transfer learning, as in the works by Feng & Yu (2020); Kang et al. (2022a) for overlapping training samples and in the works by Liu et al. (2020); Feng et al. (2022) for handling missing features. Knowledge distillation is another approach, as used by Ren et al. (2022); Li et al. (2023b) for overlapping samples and by Li et al. (2023a); Huang et al. (2023) which leverage nonaligned samples when training local predictors (only the latter considers scenarios with more than two clients). Xiao et al. (2024) recently proposed using a distributed generative adversarial network for collaborative training on nonoverlapping data and synthetic data generation. These methods yield local predictors that outperform naive local approaches, but fail to utilize valuable information when other feature blocks are available during inference.

A few recent works enable a varying number of clients to collaborate during inference. Sun et al. (2024) employ party-wise dropout during training to mitigate performance drops from missing feature blocks at inference; Gao et al. (2024) introduce a multi-stage approach with complementary knowledge distillation, enabling inference with different subsets of clients for binary classification tasks; and Ganguli et al. (2024) extend Sun et al. (2024) to deal with communication failures during inference in cross-device VFL. However, all of these methods consider the case of fully-observed training data and they all lack convergence guarantees.

In contrast to prior work, `LASER-VFL` can handle any subset of feature blocks being present during both training and inference without wasting data, while requiring only minor modifications to the standard VFL approach. It differs from conventional VFL solely in its use of parameter-sharing and task-sampling mechanisms, without the need for additional stages.

## 2 DEFINITIONS AND PRELIMINARIES

Our global dataset $\mathcal{D}$ is drawn from an input space $\mathcal{X}$ which is partitioned into $K$ feature spaces $\mathcal{X} = \mathcal{X}_1 \times \cdots \times \mathcal{X}_K$ such that $\mathcal{D}_k \subseteq \mathcal{X}_k$, where, recall, $\mathcal{D}_k$ is the local dataset of client $k$. Ideally, we

would train a general, unconstrained predictor $\tilde{h} \colon \mathcal{X} \mapsto \mathcal{Y}$, where $\mathcal{Y}$ is the label space, by solving $\min_{\tilde{\boldsymbol{\theta}}} \frac{1}{N} \sum_{n=1}^{N} \ell(\tilde{h}(\boldsymbol{x}^n; \tilde{\boldsymbol{\theta}}), y^n)$. However, VFL brings the additional constraint that this must be achieved without sharing the local datasets. That is, for all $k$, the local dataset $\mathcal{D}_k$ must remain at client $k$. As mentioned in Section 1, standard VFL methods approximate $\tilde{h}$ by $h \colon \mathcal{X} \mapsto \mathcal{Y}$, as defined in (1), allowing for collaborative training without sharing the local datasets, but they cannot handle missing features during training nor inference.

Another way to train predictors without sharing local data is to learn local predictors. In particular, each client $k \in \mathcal{K}$ can learn the parameters $\boldsymbol{\theta}_k$ of a predictor $h_k \colon \mathcal{X}_k \mapsto \mathcal{Y}$ to approximate $\tilde{h}$:

$$h_k(\boldsymbol{x}_k^n; \boldsymbol{\theta}_k) \coloneqq g_k(\boldsymbol{f}_k(\boldsymbol{x}_k^n; \boldsymbol{\theta}_{\boldsymbol{f}_k}); \boldsymbol{\theta}_{g_k}) \quad \text{where} \quad \boldsymbol{\theta}_k \coloneqq (\boldsymbol{\theta}_{\boldsymbol{f}_k}, \boldsymbol{\theta}_{g_k}). \tag{2}$$

The representation models $\boldsymbol{f}_k \colon \mathcal{X}_k \mapsto \mathcal{E}_k$, where $\mathcal{E}_k$ is the representation space of client $k$, are as in (1), yet the fusion models $g_k \colon \mathcal{E}_k \mapsto \mathbb{R}$ differ from $g \colon \mathcal{E}_1 \times \cdots \times \mathcal{E}_K \mapsto \mathbb{R}$ in (1) and $h_k$ differs from $h$. This approach is useful in that $h_k$ allows client $k$ to perform inference (and be trained) independently of other clients, but it does not make use of the features observed by clients $j \neq k$.

More generally, to achieve robustness to missing blocks in the test data while avoiding wasting other features, we wish to be able to perform inference based on any possible subset of blocks, $\mathscr{P}(\mathcal{K}) \backslash \{\varnothing\}$, where $\mathscr{P}(\mathcal{K})$ denotes the power set of $\mathcal{K}$. Standard VFL allows us to obtain a predictor for the blocks $\mathcal{K}$ and the local approach provides us with predictors for the singletons $\{\{i\} \colon i \in \mathcal{K}\}$ in the power set. However, none of the other subsets of $\mathcal{K}$ is covered by either approach.

A naive way to achieve this would be to train a predictor for each set $\mathcal{I} \in \mathscr{P}(\mathcal{K}) \setminus \{\varnothing\}$ in a decoupled manner. That is, we could train each of the following predictors $\{h_\mathcal{I} \colon \prod_{k \in \mathcal{I}} \mathcal{X}_k \mapsto \mathcal{Y}\}$ independently using the collaborative training approach of standard VFL:

$$\left\{ h_\mathcal{I}(\boldsymbol{x}_\mathcal{I}^n; \boldsymbol{\theta}_\mathcal{I}) \coloneqq g_\mathcal{I}((\boldsymbol{f}_k(\boldsymbol{x}_k^n; \boldsymbol{\theta}_{\boldsymbol{f}_k(\mathcal{I})}) \colon k \in \mathcal{I}); \boldsymbol{\theta}_{g_\mathcal{I}}) \colon \mathcal{I} \in \mathscr{P}(\mathcal{K}) \setminus \{\varnothing\} \right\}, \tag{3}$$

where $\boldsymbol{\theta}_\mathcal{I} = ((\boldsymbol{\theta}_{\boldsymbol{f}_k(\mathcal{I})} \colon k \in \mathcal{I}), \boldsymbol{\theta}_{g_\mathcal{I}})$ and $\boldsymbol{x}_\mathcal{I}^n = (\boldsymbol{x}_k^n \colon k \in \mathcal{I})$. The fusion models $\{g_\mathcal{I}\}$ can either be at one of the clients or at the server. The set of predictors in (3) includes the local predictors $\{h_k\}$ in (2) and the standard VFL predictor $h$ in (1), but also all the other nonempty sets in the power set $\mathscr{P}(\mathcal{K})$.[2] This approach addresses the issue of limited flexibility and robustness in prior methods, which struggle with varying numbers of available or participating clients during inference. However, by requiring the independent training of $2^K - 1$ distinct predictors, it introduces a new challenge: the number of models would grow exponentially with the number of clients, $K$. Consequently, the associated memory, computation, and communication costs would also increase exponentially.

Further, if the fusion models in (3) are all held by a single entity, this setup introduces a dependency of all clients on that entity. To enhance robustness against clients dropping from the federation, we would like each client to be able to ensure inference whenever it has access to its corresponding block of the sample. That is, we want each client $k \in \mathcal{K}$ to train a predictor for every nonempty subset of $\mathcal{K}$ that includes $k$. We define this family of subsets as $\mathscr{P}_k(\mathcal{K}) \coloneqq \{\mathcal{I} \in \mathscr{P}(\mathcal{K}) \colon k \in \mathcal{I}\}$. This allows client $k$ to make predictions even if the information from blocks observed by other clients is not available and thus cannot be leveraged to improve performance.

In the next section, we present our method, `LASER-VFL`, which enables the efficient training of a family of predictors that can handle scenarios where features from an arbitrary set of clients are missing. We have included a table of notation and a diagram illustrating our method in Appendix B.

## 3 OUR METHOD

**Key idea.** The main challenge, when striving for each client to be able to perform inference from any subset of blocks that includes its own, is to circumvent the exponential complexity arising from the combinatorial explosion of possible combinations of blocks. To address this, in `LASER-VFL`, we *share model parameters* across the predictors leveraging different subsets of blocks and train them so that the representation models allow for good performance across the different combinations of blocks. Further, we train the fusion model at each client to handle any combination of representations that includes its own. To train the predictors on an exponential number of combinations of missing blocks while avoiding an exponential computational complexity, we employ a *sampling* mechanism during training, which allows us to essentially estimate an exponential combination of objectives with a subexponential complexity.

---

[2]The notation in (3) differs slightly from (1) and (2), which use a simpler, more specific formulation.

## 3.1 A TRACTABLE FAMILY OF PREDICTORS SHARING MODEL PARAMETERS

In our approach, each client $k \in \mathcal{K}$ resorts to a set of predictors $\{h_{k\mathcal{I}} : \mathcal{I} \in \mathscr{P}_k(\mathcal{K})\}$. Each predictor $h_{k\mathcal{I}} : \prod_{i \in \mathcal{I}} \mathcal{X}_i \mapsto \mathcal{Y}$ has the same target, but a different domain. Therefore, we say that each predictor performs a distinct *task*. We define the predictors of client $k$ as follows:

$$\left\{ h_{k\mathcal{I}} \left( \boldsymbol{x}_\mathcal{I}^n ; \boldsymbol{\theta}_{(k,\mathcal{I})} \right) \coloneqq g_k \left( \frac{1}{|\mathcal{I}|} \sum_{i \in \mathcal{I}} \boldsymbol{f}_i(\boldsymbol{x}_i^n ; \boldsymbol{\theta}_{\boldsymbol{f}_i}) ; \boldsymbol{\theta}_{g_k} \right) : \mathcal{I} \in \mathscr{P}_k(\mathcal{K}) \right\}, \tag{4}$$

where $\boldsymbol{\theta}_{(k,\mathcal{I})} = ((\boldsymbol{\theta}_{\boldsymbol{f}_i} : i \in \mathcal{I}), \boldsymbol{\theta}_{g_k})$ and $|\mathcal{I}|$ denotes the cardinality of set $\mathcal{I}$.

**Sharing model parameters.** Note that, although we can see (4) as $|\mathscr{P}_k(\mathcal{K})| = 2^{K-1}$ different predictors, the predictors are made up of different combinations of only $K$ representation models and $K$ fusion models. That is, we only require parameters $\boldsymbol{\theta} \coloneqq (\boldsymbol{\theta}_{\boldsymbol{f}_1}, \dots, \boldsymbol{\theta}_{\boldsymbol{f}_K}, \boldsymbol{\theta}_{g_1}, \dots, \boldsymbol{\theta}_{g_K})$. In particular, in contrast to (3), where the predictors used different representation models for each task, in (4), we share the representation models across predictors. Similarly to the representation models, we have a single fusion model per client.

It is also important to note that each fusion model $g_k \left( (\boldsymbol{f}_i(\boldsymbol{x}_i^n ; \boldsymbol{\theta}_{\boldsymbol{f}_i}) : i \in \mathcal{I}) ; \boldsymbol{\theta}_{g_k} \right)$ takes the specific form of $g_k \left( \frac{1}{|\mathcal{I}|} \sum_{i \in \mathcal{I}} \boldsymbol{f}_i(\boldsymbol{x}_i^n ; \boldsymbol{\theta}_{\boldsymbol{f}_i}) ; \boldsymbol{\theta}_{g_k} \right)$. By employing a nonparameterized aggregation mechanism on the extracted representations whose output does not depend on the number of aggregated representations, the same fusion model can handle different sets of representations of different sizes. In particular, we opt for an average (rather than, say, a sum) because neural networks perform better when their inputs have similar distributions (Ioffe & Szegedy, 2015). Note that the representation model can be adjusted so that using averaging instead of concatenation as the aggregation mechanism does not reduce the flexibility of the overall model, but simply shifts it from the fusion model to the representation models, as explained in Appendix A.2.

With this approach, we have avoided an exponential memory complexity by sharing the weights across an exponential number of predictors such that we have $K$ representation models and $K$ fusion models. In the next subsection, we will go over the efficient training of this family of predictors.

## 3.2 EFFICIENT TRAINING VIA TASK-SAMPLING

The family of predictors introduced in Section 3.1 can efficiently perform inference for an exponential number of tasks. We will now discuss how to train this family of predictors while avoiding exponential computation and communication complexity during the training process. The key to this approach lies in carefully sampling tasks at each gradient step of model training.

**Our optimization problem.** As noted in Section 1, we want to address missing features not only during inference, but also during training. Let $\mathcal{K}_o^n$ be the set of observed feature blocks of sample $n$, we assume that $\mathcal{K}_o \coloneqq \{\mathcal{K}_o^n : \forall n\}$ follows a random distribution. Recalling (4), we denote the loss $\ell$ of sample $n$, with label $y^n$, for the predictor of client $k$ using blocks $\mathcal{I} \in \mathscr{P}_k(\mathcal{K}_o^n)$ of $\boldsymbol{x}^n$ as follows:

$$\mathcal{L}_{nk\mathcal{I}}(\boldsymbol{\theta}) \coloneqq \ell \left( h_{k\mathcal{I}} \left( \boldsymbol{x}_\mathcal{I}^n ; \boldsymbol{\theta}_{(k,\mathcal{I})} \right), y^n \right).$$

For sample $n$ and client $k$, the (weighted) prediction loss across all observed subsets of blocks that include $k$ is denoted by:

$$\mathcal{L}_{nk}^o(\boldsymbol{\theta}) \coloneqq \sum_{\mathcal{I} \in \mathscr{P}_k(\mathcal{K}_o^n)} \frac{1}{|\mathcal{I}|} \cdot \mathcal{L}_{nk\mathcal{I}}(\boldsymbol{\theta}),$$

where superscript $o$ indicates that this loss concerns the observed dataset. The normalization by $|\mathcal{I}|$ avoids assigning a larger weight to tasks concerning a larger number of blocks, since the set $\mathcal{I}$ is used for prediction at $|\mathcal{I}|$ different clients, whose losses are combined in the loss of sample $n$:

$$\mathcal{L}_n^o(\boldsymbol{\theta}) \coloneqq \sum_{k \in \mathcal{K}_o^n} \mathcal{L}_{nk}^o(\boldsymbol{\theta}). \tag{5}$$

Now, to train the predictors in (4), we consider the following optimization problem:

$$\min_{\boldsymbol{\theta}} \{ \mathcal{L}(\boldsymbol{\theta}) \coloneqq \mathbb{E}[\mathcal{L}^o(\boldsymbol{\theta})] \} \quad \text{where} \quad \mathcal{L}^o(\boldsymbol{\theta}) \coloneqq \frac{1}{N} \sum_{n=1}^N \mathcal{L}_n^o(\boldsymbol{\theta}), \tag{6}$$

where the expectation is over the set of observed feature blocks, $\mathcal{K}_o$.

---

**Algorithm 1:** LASER-VFL Training

**Input:** initial point $\boldsymbol{\theta}^0$, training data $\mathcal{D}$.

1 **for** $t = 0, \ldots, T - 1$ **do**

2      Clients $\mathcal{K}$ sample the same mini-batch $\mathcal{B}^t$ with $\mathcal{K}_o^{(t)} = \mathcal{K}_o^n, \forall n \in \mathcal{B}^t$, via a shared seed.

3      **for** $k \in \mathcal{K}_o^{(t)}$ **in parallel do**

4          Broadcast $\boldsymbol{f}_k^t$ and receive $\{\boldsymbol{f}_i^t : i \in \mathcal{K}_o^{(t)}, i \neq k\}$.

5          Sample a task-defining set of blocks $\mathcal{S}_{kj}^t \sim \mathcal{U}(\mathscr{P}_k^j(\mathcal{K}_o^{(t)}))$ for $j = 1, \ldots, K_o^t$ .

6          Compute $\hat{\mathcal{L}}_{\mathcal{B}^t \mathcal{S}_k^t}(\boldsymbol{\theta}^t)$, as in (8), and backpropagate over fusion model $g_k$.

7          Send the derivative of $\hat{\mathcal{L}}_{\mathcal{B}^t \mathcal{S}_k^t}(\boldsymbol{\theta}^t)$ with respect to each $k \in \mathcal{K}_o^{(t)}$ and receive theirs.

8          Sum the received gradients and backpropagate over $\boldsymbol{f}_k$.

9      Update the weights $\boldsymbol{\theta}^{t+1} = \boldsymbol{\theta}^t - \eta \nabla \hat{\mathcal{L}}_{\mathcal{B}^t \mathcal{S}^t}(\boldsymbol{\theta}^t)$, where $\hat{\mathcal{L}}_{\mathcal{B}^t \mathcal{S}^t}(\boldsymbol{\theta}^t)$ is as in (7).

---

Note that, if different blocks of features have different probabilities of being observed, we can also normalize the loss with respect to the different probabilities, which can be computed during the entity alignment stage of VFL (see below), before taking the expectation in (5). We omit this here for simplicity. We assume that the data is missing completely at random (Zhou et al., 2024).

Looking at (5), we see that, while we are interested in tackling $K \times 2^{K-1}$ different tasks, which falls in the realm of multi-objective optimization and multi-task learning (Caruana, 1997), we opt for combining them into a single objective, rather than employing a specialized multi-task optimizer. This corresponds to a weighted form of unitary scalarization (Kurin et al., 2022).

**Our optimization method.** To solve (6), we employ to a gradient-based method, summarized in Algorithm 1, which we now describe. At iteration $t$, with the current iterate $\boldsymbol{\theta}^t$, we select mini-batch indices $\mathcal{B}^t \subseteq [N]$, where $[N] := \{1, \ldots, N\}$, such that the set of observed feature blocks for each sample $n \in \mathcal{B}^t$, denoted by $\mathcal{K}_o^n$, is identical. We denote this common set by $\mathcal{K}_o^{(t)}$, that is, $\mathcal{K}_o^{(t)} = \mathcal{K}_o^n$ for all $n \in \mathcal{B}^t$, and define $\mathcal{L}_{\mathcal{B}}^o(\boldsymbol{\theta}) := \frac{1}{|\mathcal{B}|} \sum_{n \in \mathcal{B}} \mathcal{L}_n^o(\boldsymbol{\theta})$. Then, each client $k \in \mathcal{K}_o^{(t)}$ broadcasts its representation

$$\boldsymbol{f}_k^t := \boldsymbol{f}_k\left(\boldsymbol{x}_k^{\mathcal{B}^t}; \boldsymbol{\theta}_{\boldsymbol{f}_k}^t\right) \quad \text{where} \quad \boldsymbol{x}_k^{\mathcal{B}^t} := \{\boldsymbol{x}_k^n : \forall n \in \mathcal{B}^t\}$$

to the other clients in $\mathcal{K}_o^{(t)}$. At this stage, each client $k \in \mathcal{K}_o^{(t)}$ holds the representations corresponding to the set of observed feature blocks of the current mini-batch, $\{\boldsymbol{f}_i^t : i \in \mathcal{K}_o^{(t)}\}$.

To train the model on all possible combinations of feature blocks without incurring exponential computational complexity at each iteration, each client $k \in \mathcal{K}_o^{(t)}$ samples $K_o^t := |\mathcal{K}_o^{(t)}|$ tasks from the $2^{\mathcal{K}_o^{(t)}-1}$ tasks in $\mathscr{P}_k(\mathcal{K}_o^{(t)})$—the subset of the power set of $\mathcal{K}_o^{(t)}$ whose elements contain $k$. These sampled tasks estimate $\mathcal{L}_{\mathcal{B}^t}^o(\boldsymbol{\theta}^t)$ without computing predictions for an exponential number of tasks (note that $K_o^t \leq K$). We use this estimate to update the model parameters.

More precisely, the task-defining feature blocks sampled by client $k$ at step $t$ are given by $\mathcal{S}_k^t := \{\mathcal{S}_{k1}^t, \ldots, \mathcal{S}_{kK_o^t}^n\}$, where set $\mathcal{S}_{ki}^t$ contains $i$ feature blocks. Each set $\mathcal{S}_{ki}^t$ is sampled from a uniform distribution over a subset of $\mathscr{P}_k(\mathcal{K}_o^{(t)})$ that contains its sets of size $i$. That is:

$$\mathcal{S}_{ki}^t \sim \mathcal{U}(\mathscr{P}_k^i(\mathcal{K}_o^{(t)})) \quad \text{where} \quad \mathscr{P}_k^i(\mathcal{K}_o^{(t)}) := \{\mathcal{I} \in \mathscr{P}_k(\mathcal{K}_o^{(t)}) : |\mathcal{I}| = i\}.$$

This task-sampling mechanism allows us to estimate the loss with linear complexity, instead of the exponential cost of exact computation. More precisely, we use the following loss estimate:

$$\hat{\mathcal{L}}_{\mathcal{B}^t \mathcal{S}^t}(\boldsymbol{\theta}^t) := \sum_{k \in \mathcal{K}_o^{(t)}} \hat{\mathcal{L}}_{\mathcal{B}^t \mathcal{S}_k^t}(\boldsymbol{\theta}^t), \tag{7}$$

where $\mathcal{S}^t := \{\mathcal{S}_1^t, \ldots, \mathcal{S}_K^t\}$ and each $\hat{\mathcal{L}}_{\mathcal{B}^t \mathcal{S}_k^t}(\boldsymbol{\theta}^t)$, computed at client $k$, is given by:

$$\hat{\mathcal{L}}_{\mathcal{B}^t \mathcal{S}_k^t}(\boldsymbol{\theta}^t) := \frac{1}{|\mathcal{B}^t|} \sum_{n \in \mathcal{B}^t} \sum_{i=1}^{K_o^t} a_i^t \cdot \mathcal{L}_{nk\mathcal{S}_{ki}^t}(\boldsymbol{\theta}^t) \quad \text{where} \quad a_i^t := \frac{1}{i} \cdot \binom{K_o^t - 1}{i - 1}. \tag{8}$$

---

**Algorithm 2:** `LASER-VFL` Inference

---

**Input:** test sample $\boldsymbol{x}'$, trained parameters $\boldsymbol{\theta}^T$.

1 The blocks of features $\mathcal{K}'_o$ of sample $\boldsymbol{x}'$ are observed.

2 **for** $k \in \mathcal{K}'_o$ **in parallel do**

3      Broadcast $\boldsymbol{f}_k(\boldsymbol{x}'_k; \boldsymbol{\theta}^T_{\boldsymbol{f}_k})$ and receive $\boldsymbol{f}_i(\boldsymbol{x}'_i; \boldsymbol{\theta}^T_{\boldsymbol{f}_i})$, for all $i \in \mathcal{K}'_o \setminus \{k\}$.

4      Client $k$ predicts the label using $h_{k\mathcal{K}'_o}\left(\boldsymbol{x}^n_{\mathcal{K}'_o}; \boldsymbol{\theta}_{(k,\mathcal{K}'_o)}\right)$.

---

The weighting $a^t_i$ counteracts the probability of each task being sampled, allowing us to obtain an unbiased estimate of the observed loss. Now, we use this to minimize (5) by doing the following update:

$$\boldsymbol{\theta}^{t+1} = \boldsymbol{\theta}^t - \eta \nabla \hat{\mathcal{L}}_{\mathcal{B}^t \mathcal{S}^t}(\boldsymbol{\theta}^t).$$

After completing the training described in Algorithm 1, we can perform inference according to Algorithm 2. In this inference algorithm, each client $k \in \mathcal{K}'_o$ that observes a partition of the test sample $\boldsymbol{x}'$ shares its corresponding representation of $\boldsymbol{x}'$. Then, each such client $k$ uses $h_{(k,\mathcal{K}'_o)}$ to predict the label.

**Block availability-aware entity alignment.** Entity alignment is a process the precedes training in VFL where the samples of the different local datasets are matched so that their features are correctly aligned, thus allowing for collaborative model training (Liu et al., 2024). The identification of the available blocks for each sample can be performed during this prelude to VFL training, allowing for a batch selection procedure that ensures that each batch contains samples with the same observed blocks, which we use in our method.

**Extensions to our method.** It is important to highlight that our method can also be easily employed in situations where different clients hold different labels, as it naturally fits multi-task learning problems. Further, it can easily be applied to setups where any subset of $\mathcal{K}$ holds the labels, sufficing to drop fusion models from the clients which do not hold the labels or have them tackle unsupervised or self-supervised learning tasks instead.

## 4 CONVERGENCE GUARANTEES

In this section, we present convergence guarantees for our method. Let us start by stating the assumptions used in our results.

**Assumption 1** (*L-smoothness and finite infimum*). *Function $\mathcal{L}^o \colon \boldsymbol{\Theta} \mapsto \mathbb{R}$ is differentiable and there exists a positive constant $L$ such that:*

$$\forall\, \boldsymbol{\theta}_1, \boldsymbol{\theta}_2 \in \boldsymbol{\Theta} \colon \quad \|\nabla \mathcal{L}^o(\boldsymbol{\theta}_1) - \nabla \mathcal{L}^o(\boldsymbol{\theta}_2)\| \le L\|\boldsymbol{\theta}_1 - \boldsymbol{\theta}_2\|. \tag{A1}$$

*The inequality above holds if $\mathcal{L}$ is $L'$-smooth. We assume and define $\mathcal{L}^\star := \inf_{\boldsymbol{\theta}} \mathcal{L}(\boldsymbol{\theta}) > -\infty$.*

**Assumption 2** (*Unbiasedness*). *The mini-batch gradient estimate is unbiased. That is, we have that:*

$$\forall \boldsymbol{\theta} \in \boldsymbol{\Theta} \colon \quad \mathbb{E}\left[\nabla \mathcal{L}^o_{\mathcal{B}}(\boldsymbol{\theta}) \mid \mathcal{K}_o\right] = \nabla \mathcal{L}^o(\boldsymbol{\theta}), \tag{A2}$$

*where the expectation is with respect to the mini-batch $\mathcal{B}$.*

**Assumption 3** (*Bounded variance*). *There exist positive constants $\sigma_b$ and $\sigma_s$ such that, for all $\boldsymbol{\theta} \in \boldsymbol{\Theta}$ and $k \in \mathcal{K}$, we have that:*

$$\mathbb{E}\left[\|\nabla \mathcal{L}^o_{\mathcal{B}}(\boldsymbol{\theta}) - \nabla \mathcal{L}^o(\boldsymbol{\theta})\|^2 \mid \mathcal{K}_o\right] \le \sigma^2_b \quad and \quad \mathbb{E}\left[\left\|\nabla \hat{\mathcal{L}}_{\mathcal{B}\mathcal{S}_k}(\boldsymbol{\theta}) - \nabla \mathcal{L}^o_{\mathcal{B}k}(\boldsymbol{\theta})\right\|^2 \mid \mathcal{K}_o, \mathcal{B}\right] \le \sigma^2_s, \tag{A3}$$

*where $\mathcal{L}^o_{\mathcal{B}k}(\boldsymbol{\theta}) := \frac{1}{|\mathcal{B}|} \sum_{n \in \mathcal{B}} \mathcal{L}^o_{nk}(\boldsymbol{\theta})$ and the (conditional) expectations are with respect to mini-batch $\mathcal{B}$ and with respect to the tasks sampled at each client $k$, $\mathcal{S}_k$, respectively.*

We use the following set in our results:

$$\mathcal{F}^t := \{\mathcal{B}^0, \mathcal{S}^0, \mathcal{B}^1, \mathcal{S}^1, \ldots, \mathcal{B}^{t-1}, \mathcal{S}^{t-1}\}.$$

Let us start by presenting a lemma which we resort to in our proof of Theorem 1.

**Lemma 1** (Unbiased update vector). *If the mini-batch gradient estimate is unbiased* (A2), *then, for all $t \geq 0$:*

$$\mathbb{E}\left[\nabla \hat{\mathcal{L}}_{\mathcal{B}^t \mathcal{S}^t}(\boldsymbol{\theta}^t) \mid \mathcal{K}_o, \mathcal{F}^t\right] = \nabla \mathcal{L}^o(\boldsymbol{\theta}^t), \tag{9}$$

*where the expectation is with respect to mini-batch $\mathcal{B}^t$ and the sampled set of tasks $\mathcal{S}^t$.*

**Theorem 1** (Main result). *Let $\{\boldsymbol{\theta}^t\}$ be a sequence generated by Algorithm 1, if $\mathcal{L}$ is $L$-smooth and has a finite infimum* (A1), *the mini-batch estimate of the gradient is unbiased* (A2), *and our update vector has a bounded variance* (A3), *we then have that, for $\eta \in (0, 1/L]$:*

$$\frac{1}{T} \sum_{t=0}^{T-1} \mathbb{E} \left\| \nabla \mathcal{L}(\boldsymbol{\theta}^t) \right\|^2 \leq \frac{2\Delta}{\eta T} + \eta L \left(\sigma_b^2 + K \cdot \sigma_s^2\right), \tag{10}$$

*where $\Delta := \mathcal{L}(\boldsymbol{\theta}^0) - \mathcal{L}^\star$ and the expectation is with respect to $\{\mathcal{B}^t\}$, $\{\mathcal{S}^t\}$, and $\mathcal{K}_o$.*

If we take the stepsize to be $\eta = \sqrt{\frac{2\Delta}{LT} \left(\sigma_b^2 + K \cdot \sigma_s^2\right)^{-1}}$, we get from (10) that

$$\frac{1}{T} \sum_{t=0}^{T-1} \mathbb{E} \left\| \nabla \mathcal{L}(\boldsymbol{\theta}^t) \right\|^2 \leq \sqrt{\frac{8\Delta L}{T} \left(\sigma_b^2 + K \cdot \sigma_s^2\right)},$$

thus achieving a $\mathcal{O}(1/\sqrt{T})$ convergence rate. Note the effect of the variance induced by the task-sampling mechanism, which grows with $K$.

We can further establish linear convergence to a neighborhood around the optimum under the Polyak-Łojasiewicz inequality (Polyak, 1963).

**Assumption 4** (PL inequality). *We assume that there exists a positive constant $\mu$ such that*

$$\forall \boldsymbol{\theta} \in \boldsymbol{\Theta}: \quad \|\nabla \mathcal{L}(\boldsymbol{\theta})\|^2 \geq 2\mu(\mathcal{L}(\boldsymbol{\theta}) - \mathcal{L}^\star). \tag{A4}$$

We use the expected suboptimality $\delta^t := \mathbb{E}\mathcal{L}(\boldsymbol{\theta}^t) - \mathcal{L}^\star$, where the expectation is with respect to $\{\mathcal{B}^t\}$, $\{\mathcal{S}^t\}$, and $\mathcal{K}_o$, as our Lyapunov function in the following result.

**Theorem 2** (Linear convergence). *Let $\{\boldsymbol{\theta}^t\}$ be a sequence generated by Algorithm 1, if $\mathcal{L}$ is $L$-smooth and has a finite infimum* (A1), *the mini-batch estimate of the gradient is unbiased* (A2), *our update vector has a bounded variance* (A3), *and the PL inequality* (A4) *holds for $\mathcal{L}$, we then have that, for $\eta \in (0, 1/L]$:*

$$\delta^T \leq (1 - \mu\eta)^T \delta^0 + \frac{\eta L}{2\mu} \left(\sigma_b^2 + K \cdot \sigma_s^2\right).$$

It follows from $\mu \leq L$ that $1 - \mu\eta \in (0, 1)$. Therefore, we have linear convergence to a $\mathcal{O}\left(\sigma_b^2 + K \cdot \sigma_s^2\right)$ neighborhood of the global optimum.

We defer the proofs of Lemma 1, Theorem 1, and Theorem 2 to Appendix C.

## 5 EXPERIMENTS

In this section, we describe our numerical experiments and analyze their results, comparing the empirical performance of `LASER-VFL` against baseline approaches. We begin with a brief overview of the baseline methods and tasks considered, followed by a the presentation and discussion of the experimental results.

We compare our method to the following baselines:

- `Local`: as in (2); the local approach leverages its block for training and inference whenever it is observed, but ignores the remaining blocks.
- `Standard VFL` (Liu et al., 2022): as in (1); the standard VFL model can only be trained on and used for inference for fully-observed samples. When the model is unavailable for prediction, a random prediction is made.
- `VFedTrans` (Huang et al., 2023): `VFedTrans` introduces a multi-stage VFL training pipeline to collaboratively train local predictors. (We give more details about VFedTrans below.)

Table 1: Test metrics for $K = 4$ clients across different datasets and varying probabilities of missing blocks during training and inference, averaged over five seeds ($\pm$ standard deviation).

| Training $p_{\text{miss}}$ | 0.0 | | | 0.1 | | | 0.5 | | |
|---|---|---|---|---|---|---|---|---|---|
| Inference $p_{\text{miss}}$ | 0.0 | 0.1 | 0.5 | 0.0 | 0.1 | 0.5 | 0.0 | 0.1 | 0.5 |
| HAPT (accuracy, %) | | | | | | | | | |
| Local | $82.3 \pm 0.4$ | $82.5 \pm 0.5$ | $82.1 \pm 1.2$ | $81.7 \pm 0.6$ | $81.8 \pm 0.4$ | $81.5 \pm 1.6$ | $80.7 \pm 0.9$ | $80.8 \pm 0.8$ | $80.5 \pm 2.1$ |
| Standard VFL | $91.6 \pm 0.9$ | $67.3 \pm 3.5$ | $16.0 \pm 4.1$ | $90.0 \pm 1.4$ | $66.0 \pm 2.8$ | $15.8 \pm 3.9$ | $84.6 \pm 2.5$ | $62.2 \pm 2.2$ | $15.3 \pm 3.5$ |
| VFedTrans | $82.5 \pm 0.4$ | $82.5 \pm 0.4$ | $82.3 \pm 0.7$ | $82.9 \pm 0.3$ | $82.9 \pm 0.2$ | $82.8 \pm 0.7$ | $82.2 \pm 0.5$ | $82.2 \pm 0.5$ | $82.2 \pm 0.8$ |
| Ensemble | $90.9 \pm 0.9$ | $90.3 \pm 0.9$ | $84.8 \pm 1.1$ | $89.8 \pm 0.4$ | $89.3 \pm 0.5$ | $84.1 \pm 1.6$ | $88.7 \pm 1.5$ | $88.1 \pm 1.8$ | $83.0 \pm 2.5$ |
| Combinatorial | $92.5 \pm 0.2$ | $92.4 \pm 0.3$ | $88.5 \pm 1.7$ | $90.0 \pm 1.6$ | $90.0 \pm 1.4$ | $\mathbf{87.3} \pm 1.6$ | $84.2 \pm 2.5$ | $84.7 \pm 1.9$ | $83.0 \pm 1.3$ |
| PlugVFL | $91.1 \pm 1.4$ | $88.5 \pm 1.6$ | $75.1 \pm 6.3$ | $90.9 \pm 0.9$ | $88.6 \pm 2.6$ | $78.3 \pm 4.8$ | $87.7 \pm 1.6$ | $87.2 \pm 1.5$ | $82.8 \pm 1.6$ |
| LASER-VFL (ours) | $\mathbf{94.2} \pm 0.1$ | $\mathbf{93.9} \pm 0.2$ | $\mathbf{89.0} \pm 1.1$ | $\mathbf{92.4} \pm 1.3$ | $\mathbf{92.0} \pm 1.3$ | $86.6 \pm 1.2$ | $\mathbf{90.1} \pm 3.2$ | $\mathbf{89.5} \pm 3.1$ | $\mathbf{84.8} \pm 2.6$ |
| Credit ($F_1$-score$\times 100$) | | | | | | | | | |
| Local | $37.7 \pm 3.1$ | $37.6 \pm 3.1$ | $37.5 \pm 3.2$ | $37.6 \pm 3.0$ | $37.6 \pm 2.9$ | $37.5 \pm 3.2$ | $36.2 \pm 3.5$ | $36.2 \pm 3.6$ | $36.0 \pm 3.6$ |
| Standard VFL | $45.7 \pm 2.6$ | $38.8 \pm 2.5$ | $31.0 \pm 0.7$ | $42.4 \pm 1.2$ | $38.3 \pm 0.7$ | $31.0 \pm 0.7$ | $32.4 \pm 2.4$ | $32.5 \pm 1.3$ | $30.3 \pm 0.5$ |
| VFedTrans | $37.7 \pm 1.5$ | $37.6 \pm 1.4$ | $37.2 \pm 1.6$ | $39.5 \pm 0.8$ | $39.4 \pm 0.8$ | $39.2 \pm 0.9$ | $35.7 \pm 0.3$ | $35.6 \pm 0.3$ | $35.6 \pm 0.4$ |
| Ensemble | $42.1 \pm 1.0$ | $42.1 \pm 1.0$ | $40.1 \pm 0.8$ | $41.4 \pm 1.3$ | $40.8 \pm 0.9$ | $39.2 \pm 1.1$ | $\mathbf{42.4} \pm 2.1$ | $\mathbf{41.8} \pm 1.7$ | $40.1 \pm 1.1$ |
| Combinatorial | $42.8 \pm 2.9$ | $42.7 \pm 2.5$ | $41.7 \pm 1.5$ | $\mathbf{44.4} \pm 1.0$ | $41.8 \pm 2.2$ | $\mathbf{41.7} \pm 1.5$ | $32.8 \pm 3.6$ | $36.3 \pm 0.6$ | $37.6 \pm 2.1$ |
| PlugVFL | $44.4 \pm 3.7$ | $41.3 \pm 4.0$ | $32.7 \pm 4.6$ | $40.8 \pm 2.0$ | $39.8 \pm 1.6$ | $37.9 \pm 1.7$ | $38.6 \pm 2.1$ | $37.8 \pm 0.9$ | $35.4 \pm 3.4$ |
| LASER-VFL (ours) | $\mathbf{46.5} \pm 2.8$ | $\mathbf{45.0} \pm 2.5$ | $\mathbf{43.7} \pm 1.2$ | $43.1 \pm 4.2$ | $\mathbf{41.9} \pm 4.0$ | $41.3 \pm 1.9$ | $41.5 \pm 4.2$ | $40.9 \pm 4.0$ | $\mathbf{41.4} \pm 1.7$ |
| MIMIC-IV ($F_1$-score$\times 100$) | | | | | | | | | |
| Local | $74.9 \pm 0.5$ | $74.9 \pm 0.5$ | $74.8 \pm 0.5$ | $75.0 \pm 0.3$ | $75.0 \pm 0.2$ | $75.0 \pm 0.3$ | $73.0 \pm 0.5$ | $73.0 \pm 0.5$ | $73.0 \pm 0.5$ |
| Standard VFL | $91.6 \pm 0.2$ | $77.0 \pm 1.3$ | $52.5 \pm 2.2$ | $90.9 \pm 0.3$ | $76.8 \pm 1.3$ | $52.5 \pm 2.2$ | $81.1 \pm 0.4$ | $70.2 \pm 0.8$ | $51.8 \pm 1.8$ |
| Ensemble | $84.2 \pm 0.3$ | $83.9 \pm 0.3$ | $77.9 \pm 1.6$ | $84.1 \pm 0.4$ | $83.7 \pm 0.5$ | $78.1 \pm 1.1$ | $82.1 \pm 0.3$ | $81.8 \pm 0.5$ | $76.4 \pm 1.1$ |
| Combinatorial | $91.3 \pm 0.3$ | $87.2 \pm 1.7$ | $83.6 \pm 0.4$ | $91.0 \pm 0.3$ | $86.7 \pm 1.4$ | $83.4 \pm 0.4$ | $80.5 \pm 1.2$ | $80.6 \pm 1.7$ | $78.9 \pm 0.5$ |
| PlugVFL | $91.2 \pm 0.5$ | $89.0 \pm 0.8$ | $79.6 \pm 2.8$ | $90.6 \pm 0.2$ | $88.8 \pm 0.5$ | $79.8 \pm 2.6$ | $86.5 \pm 0.7$ | $85.3 \pm 0.9$ | $77.9 \pm 2.8$ |
| LASER-VFL (ours) | $\mathbf{92.0} \pm 0.2$ | $\mathbf{91.2} \pm 0.3$ | $\mathbf{85.5} \pm 1.0$ | $\mathbf{91.1} \pm 0.5$ | $\mathbf{90.2} \pm 0.3$ | $\mathbf{84.7} \pm 0.9$ | $\mathbf{87.7} \pm 0.2$ | $\mathbf{86.9} \pm 0.2$ | $\mathbf{82.0} \pm 1.0$ |
| CIFAR-10 (accuracy, %) | | | | | | | | | |
| Local | $76.1 \pm 0.1$ | $76.0 \pm 0.1$ | $76.2 \pm 0.2$ | $75.6 \pm 0.1$ | $75.6 \pm 0.1$ | $75.7 \pm 0.3$ | $71.2 \pm 0.4$ | $71.1 \pm 0.4$ | $71.3 \pm 0.7$ |
| Standard VFL | $89.7 \pm 0.2$ | $60.5 \pm 12.1$ | $11.6 \pm 3.5$ | $87.0 \pm 0.6$ | $58.8 \pm 11.4$ | $11.5 \pm 3.4$ | $54.9 \pm 10.0$ | $38.6 \pm 9.1$ | $10.9 \pm 2.2$ |
| Ensemble | $86.4 \pm 0.2$ | $85.2 \pm 0.7$ | $78.3 \pm 0.9$ | $86.1 \pm 0.3$ | $85.0 \pm 0.5$ | $77.8 \pm 0.7$ | $82.4 \pm 0.2$ | $81.0 \pm 0.8$ | $73.8 \pm 0.6$ |
| Combinatorial | $89.4 \pm 0.1$ | $88.7 \pm 0.4$ | $83.4 \pm 1.2$ | $87.1 \pm 0.8$ | $86.7 \pm 0.4$ | $82.4 \pm 0.8$ | $54.9 \pm 9.7$ | $58.6 \pm 7.7$ | $68.4 \pm 3.1$ |
| PlugVFL | $89.4 \pm 0.2$ | $88.1 \pm 0.8$ | $81.8 \pm 1.7$ | $88.1 \pm 0.5$ | $86.9 \pm 0.6$ | $81.2 \pm 1.2$ | $76.5 \pm 2.1$ | $75.6 \pm 1.5$ | $72.4 \pm 1.0$ |
| LASER-VFL (ours) | $\mathbf{91.5} \pm 0.1$ | $\mathbf{90.5} \pm 0.4$ | $\mathbf{83.8} \pm 1.5$ | $\mathbf{91.2} \pm 0.2$ | $\mathbf{90.2} \pm 0.4$ | $\mathbf{83.3} \pm 1.4$ | $\mathbf{87.4} \pm 0.3$ | $\mathbf{86.4} \pm 0.6$ | $\mathbf{79.4} \pm 1.6$ |
| CIFAR-100 (accuracy, %) | | | | | | | | | |
| Local | $50.3 \pm 0.1$ | $50.3 \pm 0.1$ | $50.4 \pm 0.1$ | $49.1 \pm 0.2$ | $49.2 \pm 0.2$ | $49.1 \pm 0.3$ | $41.3 \pm 0.7$ | $41.3 \pm 0.6$ | $41.3 \pm 0.7$ |
| Standard VFL | $64.7 \pm 0.4$ | $41.5 \pm 9.7$ | $2.4 \pm 2.9$ | $57.2 \pm 2.2$ | $36.6 \pm 7.6$ | $2.3 \pm 2.8$ | $15.8 \pm 7.0$ | $10.5 \pm 4.6$ | $1.4 \pm 1.0$ |
| Ensemble | $62.1 \pm 0.3$ | $60.2 \pm 1.2$ | $52.3 \pm 0.9$ | $60.8 \pm 0.3$ | $58.8 \pm 1.0$ | $51.0 \pm 0.5$ | $52.6 \pm 0.6$ | $50.4 \pm 0.7$ | $43.5 \pm 0.8$ |
| Combinatorial | $64.4 \pm 0.5$ | $64.0 \pm 0.4$ | $58.5 \pm 1.3$ | $57.2 \pm 2.2$ | $57.8 \pm 1.8$ | $55.5 \pm 0.7$ | $16.4 \pm 6.9$ | $19.5 \pm 5.9$ | $31.6 \pm 4.3$ |
| PlugVFL | $66.0 \pm 0.3$ | $64.1 \pm 1.2$ | $55.3 \pm 2.1$ | $63.2 \pm 1.1$ | $61.5 \pm 0.7$ | $53.1 \pm 1.9$ | $44.6 \pm 2.3$ | $43.9 \pm 1.7$ | $41.2 \pm 1.6$ |
| LASER-VFL (ours) | $\mathbf{72.3} \pm 0.1$ | $\mathbf{71.0} \pm 0.7$ | $\mathbf{61.3} \pm 1.8$ | $\mathbf{70.4} \pm 0.3$ | $\mathbf{68.9} \pm 0.6$ | $\mathbf{59.8} \pm 1.6$ | $\mathbf{61.4} \pm 0.9$ | $\mathbf{59.9} \pm 0.5$ | $\mathbf{51.9} \pm 1.2$ |

- Ensemble: we train local predictors as in Local. During inference, the clients share their predictions, and a joint prediction is selected by majority vote, with ties broken at random.

- Combinatorial: as in (3); during training, each batch is used by all the models corresponding to subsets of the observed blocks. During inference, we use the predictor corresponding to the set of observed blocks. We go over the scalability problems of this approach below.

- PlugVFL (Sun et al., 2024): we extend PlugVFL to handle missing features during training by replacing missing representations with zeros. This method matches the work proposed by (Ganguli et al., 2024) for scenarios where all clients have reliable links to one another.

Our experiments focus on the following tasks:

- **HAPT** (Reyes-Ortiz et al., 2016): a human activity recognition dataset.

- **Credit** (Yeh & Lien, 2009): a dataset for predicting credit card payment defaults.

- **MIMIC-IV** (Johnson et al., 2020): a time series dataset concerning healthcare data. We focus on the task of predicting in-hospital mortality from ICU data corresponding to patients admitted with chronic kidney disease. We resort to the data processing pipeline of Gupta et al. (2022).

- **CIFAR-10 & CIFAR-100** (Krizhevsky et al., 2009): two popular image datasets.

For all datasets, we split the samples into $K = 4$ feature blocks. For example, for the CIFAR datasets, each image is partitioned into four quadrants, each assigned to a different client. Given our interest in ensuring that all clients perform well at inference time, we average the metrics across clients. In Appendix D, we provide detailed descriptions of how we compute accuracy and $F_1$-score.

Experiments on HAPT and Credit show that VFedTrans does not significantly outperform Local despite being considerably more complex and slower. The training pipeline of VFedTrans includes three stages: federated representation learning, local representation distillation, and training a local model. The first two stages are repeated $K - 1$ times for each client pair. This makes VFedTrans time-intensive to run and hyperparameter tune, as seen in Appendix D. For this reason, we limit our experiments with VFedTrans to these simpler tasks.

**Performance for different missing data probabilities.** In Table 1, we see that the methods with local predictors (`Local` and `VFedTrans`) are robust to missing blocks during both training and inference, yet they fall behind when all the blocks are observed. In contrast, `Standard VFL` performs well when all the blocks are observed, yet its performance degrades very quickly in the presence of missing blocks. The `Ensemble` method improves upon `Standard VFL` significantly in the presence of missing blocks, as it leverages the robustness its local predictors. Yet, when all the blocks are observed, although `Ensemble` improves over the local predictors significantly, it still cannot match the performance of `Standard VFL`. We see that the `Combinatorial` approach outperforms most baselines when there is little to no missing training data ($p_{\text{miss}} \in \{0.0, 0.1\}$). Yet, for a larger amount of missing training data ($p_{\text{miss}} = 0.5$), its performance degrades significantly. This degradation is due to the fact that each mini-batch is only used to train predictors that use observed feature blocks, harming the generalization of predictors that use a larger subset of the features.

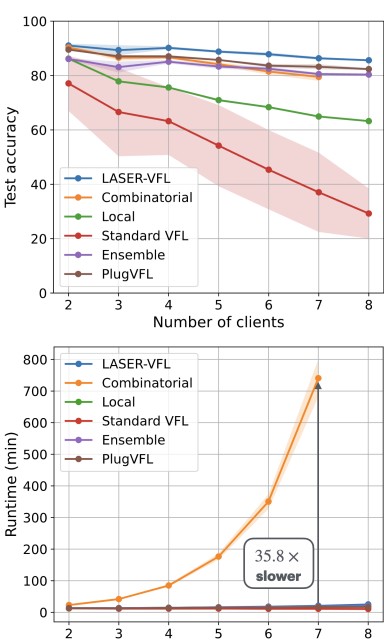

This also leads to an interesting phenomenon: when $p_{\text{miss}} = 0.5$ for training data, `Combinatorial` performs better for higher probabilities of missing testing data. This is because missing test data dictates the use of predictors trained on smaller, more frequently observed feature subsets. `PlugVFL` often outperforms most of the other baselines, though not consistently. Specifically, it is at times outperformed by `Combinatorial` when the missing probability of the training data is low, and by `Ensemble` when it is high. `LASER-VFL` consistently outperforms the baselines across the different probabilities of missing blocks during training and inference. (In fact, in Table 1, `LASER-VFL` is never outperformed by more than a standard deviation.) In particular, even when the samples are fully-observed, `LASER-VFL` outperforms `Standard VFL` and `Combinatorial`. We believe this is due to the regularization effect of the task-sampling mechanism in our method, which effectively behaves as a form of dropout.

**Scalability.** Figure 2 examines the scalability of the different methods with respect to the number of clients, $K$. In the top plot, we see that `LASER-VFL` performs the best for all $K$. It is followed by `Combinatorial`, `Ensemble`, and `PlugVFL`. However, the performance of `Combinatorial` drops faster as $K$ increases, since each exact combination of feature blocks is observed less frequently. In contrast, `Ensemble` and `PlugVFL`, like `LASER-VFL`, train each representation model on all batches for which its feature block is observed. In the bottom plot, we observe the expected scalability issues of `Combinatorial`, caused by its exponential complexity.

Figure 2: Performance and runtime across different numbers of clients (CIFAR-10, $p_{\text{miss}} = 0.1$ for training and inference). We did not run `Combinatorial` for 8 clients due to resource constraints.

**Nonuniform missing features.** Table 3 in Appendix E presents an experiment where the probability of each feature block $k$ not being observed is independent and identically distributed as $p_{\text{miss}}(k) \sim \text{Beta}(2.0, 2.0)$. As in the previous experiments, `LASER-VFL` outperforms the baselines.

## 6 CONCLUSIONS

In this work, we introduced `LASER-VFL`, a novel method for efficient training and inference in vertical federated learning that is robust to missing blocks of features without wasting data. This is achieved by carefully sharing model parameters and by employing a task-sampling mechanism that allows us to efficiently estimate a loss whose computation would otherwise lead to an exponential complexity. We provide convergence guarantees for `LASER-VFL` and present numerical experiments demonstrating its improved performance, not only in the presence of missing features but, remarkably, even when all samples are fully observed. For future work, it would be interesting to apply `LASER-VFL` to multi-task learning and investigate how it behaves when different clients tackle different tasks—a scenario that naturally fits our setup.

ACKNOWLEDGMENTS

This work is supported in part by the Fundação para a Ciência e a Tecnologia through the Carnegie Mellon Portugal Program under grant SFRH/BD/150738/2020 and by the grants U.S. National Science Foundation CCF-2007911 and ECCS-2318441.

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

# A  ADDITIONAL DISCUSSIONS

## A.1  A MORE DETAILED COMPARISON WITH CLOSEST RELATED WORKS

As mentioned in Section 1, a few recent works have addressed the problem of missing features during VFL training. In particular, a line of work uses all samples that are not fully observed locally to train the representation models via self-supervised learning (Feng, 2022; He et al., 2024). This is followed by collaborative training of a joint predictor for the whole federation using the samples without missing features. Another line of work utilizes a similar framework but resorting to semi-supervised learning instead of self-supervised learning in order to take advantage of nonaligned samples (Kang et al., 2022b; Yang et al., 2022; Sun et al., 2023). In both cases, these works are lacking in some areas that are improved by our method: **1)** they require the existence of fully observed samples; **2)** they do not allow for missing features at inference time; and **3)** they do not leverage collaborative training when more than one, but less than the total number of clients observed their feature blocks.

Another approach leverages information across the entire federation to train local predictors, thereby enabling inference in the presence of missing features. Feng & Yu (2020); Kang et al. (2022a) propose following standard, collaborative vertical FL with a transfer learning method allowing each client to have its own predictor. These works assume that the training samples are fully observed. Liu et al. (2020); Feng et al. (2022) follow a similar direction but they also consider the presence of missing features. Instead of utilizing transfer learning to train local predictors, Ren et al. (2022); Li et al. (2023b) use knowledge distillation under the assumption of fully observed training samples, while Li et al. (2023a); Huang et al. (2023) also employ knowledge distillation, but on top of this they further consider missing features during training. Finally, Xiao et al. (2024) recently proposed using a distributed generative adversarial network for collaborative training on nonoverlapping data, leveraging synthetic data generation. In all of these works, the output of training are local predictors which, despite being able to perform inference when other clients in the federation do not observe their blocks, are also not able to leverage other feature blocks when they *are* observed. This failure to utilize valuable information during inference leads to reduced predictive power.

Only a few, very recent works allow for inference from a varying number of feature blocks in vertical FL. `PlugVFL` (Sun et al., 2024) employs party-wise dropout during training to mitigate performance drops from missing feature blocks at inference; Gao et al. (2024) introduce a multi-stage approach with complementary knowledge distillation, enabling inference with different client subsets; and Ganguli et al. (2024) deal with the related task of handling communication failure during inference in cross-device VFL. These methods make progress towards inference in vertical FL in the presence of missing feature blocks, however, they do not consider missing features in the training data. Further, none of these methods provides convergence guarantees.

In contrast to the prior art, our `LASER-VFL` method can handle any missingness pattern in the feature blocks during both training and inference simultaneously *without wasting any data*. Further, `LASER-VFL` achieves this without requiring a complex pipeline with additional stages and hyperparameters, requiring only minor modifications to the standard VFL approach.

### A.2 ON THE USE OF AVERAGING AS THE AGGREGATION MECHANISM

As mentioned in Section 3.1, using averaging instead of concatenation as the aggregation mechanism in the fusion model does not reduce the flexibility of the overall approach, provided the representation models are adjusted accordingly. We illustrate this with a simple example employing aggregation by sum (which differs only by a parameter scaling factor).

Consider the scenario with two clients, each extracting a representation, $v_1 \in \mathbb{R}^{d_1}$ and $v_2 \in \mathbb{R}^{d_2}$. When aggregation is by concatenation, the first layer of the fusion model can be written as $W \in \mathbb{R}^{d_0 \times (d_1 + d_2)}$, with $W = [W_1 \quad W_2]$, and we define $v := \begin{bmatrix} v_1^\top & v_2^\top \end{bmatrix}^\top$. Hence, the output of the first layer is $Wv = W_1 v_1 + W_2 v_2$.

In contrast, a naive aggregation by sum would restricts us to $W_1(v_1 + v_2)$ and impose $d_1 = d_2$. However, if we include an extra linear layer in the representation models for each client (namely $W_1$ in the first and $W_2$ in the second), then the outputs become $W_1 v_1$ and $W_2 v_2$. This way, even with aggregation by summation, the fusion model can recover $W_1 v_1 + W_2 v_2$.

Thus, by adjusting the representation models appropriately, we can indeed achieve the same flexibility as in the concatenation-based approach, while using aggregation by sum or average.

## B LASER-VFL DIAGRAM AND TABLE OF NOTATION

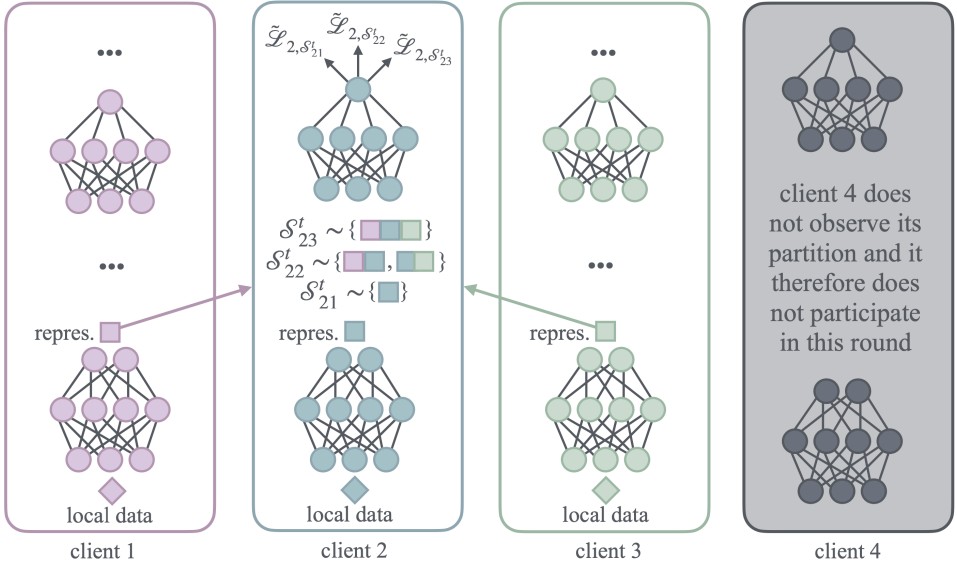

Figure 3: Diagram illustrating a forward pass of `LASER-VFL`.

Table 2: Table of Notation

| Symbol | Description |
|--------|-------------|
| $K$ | Number of clients |
| $N$ | Number of samples |
| $\boldsymbol{x}^n$ | Sample $n$ |
| $\boldsymbol{x}_k^n$ | Block $k$ of sample $n$ |
| $\mathcal{D}$ | Global dataset |
| $\mathcal{D}_k$ | Local dataset |
| $\boldsymbol{\theta}$ | Parameters of our model |
| $\boldsymbol{\theta}_{\mathcal{K}}$ | Parameters of the standard VFL model |
| $\boldsymbol{\theta}_{\boldsymbol{f}_k}$ | Parameters of the representation model at client $k$ |
| $\boldsymbol{\theta}_{\boldsymbol{g}_k}$ | Parameters of the fusion model at client $k$ |
| $\ell$ | Loss function |
| $\mathcal{L}$ | Objective function |
| $h$ | Predictor of the standard VFL model |
| $h_{(k,\mathcal{J})}$ | Predictor at client $k$ using feature blocks $\mathcal{J}$ |
| $\mathscr{P}(\mathcal{S})$ | Power set of $\mathcal{S}$ |
| $\mathscr{P}_j(\mathcal{S})$ | Subsets in the power set of $\mathcal{S}$ containing $j$ |
| $\mathscr{P}_j^i(\mathcal{S})$ | Subsets in the power set of $\mathcal{S}$ of size $i$ containing $j$ |
| $\boldsymbol{f}_k$ | Representation model at client $k$ |
| $g_k$ | Fusion model at client $k$ |
| $h_k$ | Predictor at client $k$ |
| $g$ | Standard VFL fusion model |
| $h$ | Standard VFL predictor |
| $\mathcal{K}_o^n$ | Set of observed features blocks of sample $n$ |
| $\boldsymbol{x}'$ | Test sample |

## C  PROOFS

### C.1  PRELIMINARIES

The following quadratic upper bound follows from the $L$-smoothness of $\mathcal{L}^o$ (A1):

$$\forall\, \boldsymbol{\theta}_1, \boldsymbol{\theta}_2 \in \boldsymbol{\Theta}: \quad \mathcal{L}^o(\boldsymbol{\theta}_1) \leq \mathcal{L}^o(\boldsymbol{\theta}_2) + \nabla\mathcal{L}^o(\boldsymbol{\theta}_2)^\top(\boldsymbol{\theta}_1 - \boldsymbol{\theta}_2) + \frac{L}{2}\|\boldsymbol{\theta}_1 - \boldsymbol{\theta}_2\|^2. \tag{11}$$

Let $X$ denote a random variable and let $F$ be a convex function, Jensen's inequality states that

$$F(\mathbb{E}(X)) \leq \mathbb{E}(F(X)). \tag{12}$$

We define the following shorthand notation for conditional expectations used in our proofs:

$$\mathbb{E}_o\left[\cdot\right] := \mathbb{E}\left[\cdot \mid \mathcal{K}_o\right],$$

$$\mathbb{E}_t\left[\cdot\right] := \mathbb{E}\left[\cdot \mid \mathcal{K}_o, \mathcal{F}^t\right],$$

$$\mathbb{E}_{t^+}\left[\cdot\right] := \mathbb{E}\left[\cdot \mid \mathcal{K}_o, \mathcal{F}^t, \mathcal{B}^t\right].$$

## C.2 PROOF OF LEMMA 1

From the law of total expectation, we have that

$$\mathbb{E}_t\left[\hat{\mathcal{L}}_{\mathcal{B}^t\mathcal{S}^t}(\boldsymbol{\theta}^t)\right] = \mathbb{E}_t\left[\mathbb{E}_{t^+}\left[\hat{\mathcal{L}}_{\mathcal{B}^t\mathcal{S}^t}(\boldsymbol{\theta}^t)\right]\right]. \tag{13}$$

Focusing on the inner expectation, we have from the definition of $\hat{\mathcal{L}}_{\mathcal{B}^t\mathcal{S}^t}(\boldsymbol{\theta}^t)$ in (7) and (8) that

$$\mathbb{E}_{t^+}\left[\hat{\mathcal{L}}_{\mathcal{B}^t\mathcal{S}^t}(\boldsymbol{\theta}^t)\right] = \frac{1}{|\mathcal{B}^t|}\sum_{n\in\mathcal{B}^t}\sum_{k\in\mathcal{K}_o^{(t)}}\sum_{i=1}^{K_o^t}a_i^t\cdot\mathbb{E}_{t^+}\left[\mathcal{L}_{nk\mathcal{S}_{ki}^t}(\boldsymbol{\theta}^t)\right]. \tag{14}$$

Now, from the fact that $\mathcal{S}_{ki}^t\sim\mathcal{U}(\mathscr{P}_k^i(\mathcal{K}_o^{(t)}))$, we have that

$$\begin{aligned}
\mathbb{E}_{t^+}\left[\mathcal{L}_{nk\mathcal{S}_{ki}^t}(\boldsymbol{\theta}^t)\right] &= \sum_{\mathcal{I}\in\mathscr{P}_k^i(\mathcal{K}_o^{(t)})}\mathbb{P}(\mathcal{S}_{ki}^t=\mathcal{I})\cdot\mathcal{L}_{nk\mathcal{I}}(\boldsymbol{\theta}^t)\\
&= \frac{1}{|\mathscr{P}_k^i(\mathcal{K}_o^{(t)})|}\cdot\sum_{\mathcal{I}\in\mathscr{P}_k^i(\mathcal{K}_o^{(t)})}\mathcal{L}_{nk\mathcal{I}}(\boldsymbol{\theta}^t)\\
&= \binom{K_o^t-1}{j-1}^{-1}\cdot\sum_{\mathcal{I}\in\mathscr{P}_k^i(\mathcal{K}_o^{(t)})}\mathcal{L}_{nk\mathcal{I}}(\boldsymbol{\theta}^t).
\end{aligned}$$

So, we have from (14) and from the fact that $a_i^t=\binom{K_o^t-1}{i-1}/i$, as defined in (8), that

$$\begin{aligned}
\mathbb{E}_{t^+}\left[\hat{\mathcal{L}}_{\mathcal{B}^t\mathcal{S}^t}(\boldsymbol{\theta}^t)\right] &= \frac{1}{|\mathcal{B}^t|}\sum_{n\in\mathcal{B}^t}\sum_{k\in\mathcal{K}_o^{(t)}}\sum_{i=1}^{K_o^t}\frac{1}{i}\cdot\sum_{\mathcal{I}\in\mathscr{P}_k^i(\mathcal{K}_o^{(t)})}\mathcal{L}_{nk\mathcal{I}}(\boldsymbol{\theta}^t)\\
&= \frac{1}{|\mathcal{B}^t|}\sum_{n\in\mathcal{B}^t}\sum_{k\in\mathcal{K}_o^{(t)}}\sum_{\mathcal{I}\in\mathscr{P}_k(\mathcal{K}_o^{(t)})}\frac{1}{|\mathcal{I}|}\cdot\mathcal{L}_{nk\mathcal{I}}(\boldsymbol{\theta}^t)\\
&= \mathcal{L}_{\mathcal{B}^t}^o(\boldsymbol{\theta}^t).
\end{aligned}$$

So, from (13) and (A2), it follows that

$$\mathbb{E}_t\left[\hat{\mathcal{L}}_{\mathcal{B}^t\mathcal{S}^t}(\boldsymbol{\theta}^t)\right] = \mathbb{E}_t\left[\mathcal{L}_{\mathcal{B}^t}^o(\boldsymbol{\theta}^t)\right] = \mathcal{L}^o(\boldsymbol{\theta}^t).$$

Therefore, we have from the dominated convergence theorem, which allows us to interchange the expectation and the gradient operators, that

$$\mathbb{E}_t\left[\nabla\hat{\mathcal{L}}_{\mathcal{B}^t\mathcal{S}^t}(\boldsymbol{\theta}^t)\right] = \nabla\mathbb{E}_t\left[\hat{\mathcal{L}}_{\mathcal{B}^t\mathcal{S}^t}(\boldsymbol{\theta}^t)\right] = \nabla\mathcal{L}^o(\boldsymbol{\theta}^t).$$

## C.3 PROOF OF THEOREM 1

Let us show the convergence of our method. It follows from the quadratic upper bound in (11) which ensues from the $L$-smoothness assumption (A1), that

$$\mathcal{L}^o(\boldsymbol{\theta}^{t+1}) - \mathcal{L}^o(\boldsymbol{\theta}^t) \leq \nabla\mathcal{L}^o(\boldsymbol{\theta}^t)^\top(\boldsymbol{\theta}^{t+1}-\boldsymbol{\theta}^t) + \frac{L}{2}\left\|\boldsymbol{\theta}^{t+1}-\boldsymbol{\theta}^t\right\|^2.$$

Using the fact that $\boldsymbol{\theta}^{t+1}=\boldsymbol{\theta}^t-\eta\nabla\hat{\mathcal{L}}_{\mathcal{B}^t\mathcal{S}^t}(\boldsymbol{\theta}^t)$, we get that:

$$\mathcal{L}^o(\boldsymbol{\theta}^{t+1}) - \mathcal{L}^o(\boldsymbol{\theta}^t) \leq -\eta\nabla\mathcal{L}^o(\boldsymbol{\theta}^t)^\top\nabla\hat{\mathcal{L}}_{\mathcal{B}^t\mathcal{S}^t}(\boldsymbol{\theta}^t) + \frac{\eta^2L}{2}\left\|\nabla\hat{\mathcal{L}}_{\mathcal{B}^t\mathcal{S}^t}(\boldsymbol{\theta}^t)\right\|^2.$$

We now take the conditional expectation $\mathbb{E}_t\left[\cdot\right]$, defined in Appendix C.2, arriving that:

$$\mathbb{E}_t\left[\mathcal{L}^o(\boldsymbol{\theta}^{t+1})\right] - \mathcal{L}^o(\boldsymbol{\theta}^t) \leq -\eta\nabla\mathcal{L}^o(\boldsymbol{\theta}^t)^\top\mathbb{E}_t\left[\nabla\hat{\mathcal{L}}_{\mathcal{B}^t\mathcal{S}^t}(\boldsymbol{\theta}^t)\right] + \frac{\eta^2L}{2}\mathbb{E}_t\left[\left\|\nabla\hat{\mathcal{L}}_{\mathcal{B}^t\mathcal{S}^t}(\boldsymbol{\theta}^t)\right\|^2\right].$$

Now, under the unbiasedness assumption (A2), we can use Lemma 1 to arrive at:

$$\mathbb{E}_t\left[\mathcal{L}^o(\boldsymbol{\theta}^{t+1})\right] - \mathcal{L}^o(\boldsymbol{\theta}^t) \leq -\eta\left\|\nabla\mathcal{L}^o(\boldsymbol{\theta}^t)\right\|^2 + \frac{\eta^2 L}{2}\mathbb{E}_t\left[\left\|\nabla\hat{\mathcal{L}}_{\mathcal{B}^t\mathcal{S}^t}(\boldsymbol{\theta}^t)\right\|^2\right]. \tag{15}$$

From Lemma 1, we also have that the following equation holds

$$\mathbb{E}_t\left[\left\|\nabla\hat{\mathcal{L}}_{\mathcal{B}^t\mathcal{S}^t}(\boldsymbol{\theta}^t) - \nabla\mathcal{L}^o(\boldsymbol{\theta}^t)\right\|^2\right] = \mathbb{E}_t\left[\left\|\nabla\hat{\mathcal{L}}_{\mathcal{B}^t\mathcal{S}^t}(\boldsymbol{\theta}^t)\right\|^2\right] - \left\|\nabla\mathcal{L}^o(\boldsymbol{\theta}^t)\right\|^2. \tag{16}$$

Further, we have that

$$\mathbb{E}_t\left\|\nabla\hat{\mathcal{L}}_{\mathcal{B}^t\mathcal{S}^t}(\boldsymbol{\theta}^t) - \nabla\mathcal{L}^o(\boldsymbol{\theta}^t)\right\|^2 = \mathbb{E}_t\left\|\nabla\hat{\mathcal{L}}_{\mathcal{B}^t\mathcal{S}^t}(\boldsymbol{\theta}^t) - \nabla\mathcal{L}^o_{\mathcal{B}^t}(\boldsymbol{\theta}^t) + \nabla\mathcal{L}^o_{\mathcal{B}^t}(\boldsymbol{\theta}^t) - \nabla\mathcal{L}^o(\boldsymbol{\theta}^t)\right\|^2$$

$$= \mathbb{E}_t\left\|\nabla\hat{\mathcal{L}}_{\mathcal{B}^t\mathcal{S}^t}(\boldsymbol{\theta}^t) - \nabla\mathcal{L}^o_{\mathcal{B}^t}(\boldsymbol{\theta}^t)\right\|^2 + \mathbb{E}_t\left\|\nabla\mathcal{L}^o_{\mathcal{B}^t}(\boldsymbol{\theta}^t) - \nabla\mathcal{L}^o(\boldsymbol{\theta}^t)\right\|^2$$

$$+ 2\mathbb{E}_t\left[\mathbb{E}_{t^+}\left[\left(\nabla\hat{\mathcal{L}}_{\mathcal{B}^t\mathcal{S}^t}(\boldsymbol{\theta}^t) - \nabla\mathcal{L}^o_{\mathcal{B}^t}(\boldsymbol{\theta}^t)\right)^\top\left(\nabla\mathcal{L}^o_{\mathcal{B}^t}(\boldsymbol{\theta}^t) - \nabla\mathcal{L}^o(\boldsymbol{\theta}^t)\right)\right]\right].$$

Recalling that $\mathbb{E}_{t^+}\left[\nabla\hat{\mathcal{L}}_{\mathcal{B}^t\mathcal{S}^t}(\boldsymbol{\theta}^t)\right] = \nabla\mathcal{L}^o_{\mathcal{B}^t}(\boldsymbol{\theta}^t)$, as seen in Appendix C.2, we get

$$\mathbb{E}_t\left[\mathbb{E}_{t^+}\left[\left(\nabla\hat{\mathcal{L}}_{\mathcal{B}^t\mathcal{S}^t}(\boldsymbol{\theta}^t) - \nabla\mathcal{L}^o_{\mathcal{B}^t}(\boldsymbol{\theta}^t)\right)^\top\left(\nabla\mathcal{L}^o_{\mathcal{B}^t}(\boldsymbol{\theta}^t) - \nabla\mathcal{L}^o(\boldsymbol{\theta}^t)\right)\right]\right] = 0.$$

Further, it follows from the bounded variance assumption in (A3) that

$$\mathbb{E}_t\left\|\nabla\mathcal{L}^o_{\mathcal{B}^t}(\boldsymbol{\theta}^t) - \nabla\mathcal{L}^o(\boldsymbol{\theta}^t)\right\|^2 \leq \sigma_b^2$$

and

$$\mathbb{E}_t\left\|\nabla\hat{\mathcal{L}}_{\mathcal{B}^t\mathcal{S}^t}(\boldsymbol{\theta}^t) - \nabla\mathcal{L}^o_{\mathcal{B}^t}(\boldsymbol{\theta}^t)\right\|^2 = \mathbb{E}_t\left\|\sum_{k\in\mathcal{K}_o^{\mathcal{B}^t}}\hat{\mathcal{L}}_{\mathcal{B}^t\mathcal{S}_k^t}(\boldsymbol{\theta}^t) - \nabla\mathcal{L}^o_{\mathcal{B}k}(\boldsymbol{\theta}^t)\right\|^2 \leq K\cdot\sigma_s^2.$$

Therefore, we arrive at

$$\mathbb{E}_t\left\|\nabla\hat{\mathcal{L}}_{\mathcal{B}^t\mathcal{S}^t}(\boldsymbol{\theta}^t) - \nabla\mathcal{L}^o(\boldsymbol{\theta}^t)\right\|^2 \leq \sigma_b^2 + K\cdot\sigma_s^2.$$

Using the inequality above in (16) we get that

$$\mathbb{E}_t\left[\left\|\nabla\hat{\mathcal{L}}_{\mathcal{B}^t\mathcal{S}^t}(\boldsymbol{\theta}^t)\right\|^2\right] \leq \left\|\nabla\mathcal{L}^o(\boldsymbol{\theta}^t)\right\|^2 + \sigma_b^2 + K\cdot\sigma_s^2,$$

which we can use in (15) to arrive at

$$\mathbb{E}_t\left[\mathcal{L}^o(\boldsymbol{\theta}^{t+1})\right] - \mathcal{L}^o(\boldsymbol{\theta}^t) \leq -\eta\left\|\nabla\mathcal{L}^o(\boldsymbol{\theta}^t)\right\|^2 + \frac{\eta^2 L}{2}\left\|\nabla\mathcal{L}^o(\boldsymbol{\theta}^t)\right\|^2 + \frac{\eta^2 L}{2}\left(\sigma_b^2 + K\cdot\sigma_s^2\right)$$

$$= -\eta\left(1 - \frac{\eta L}{2}\right)\left\|\nabla\mathcal{L}^o(\boldsymbol{\theta}^t)\right\|^2 + \frac{\eta^2 L}{2}\left(\sigma_b^2 + K\cdot\sigma_s^2\right).$$

Therefore, for a stepsize $\eta\in(0, 1/L]$, we have that

$$\mathbb{E}_t\left[\mathcal{L}^o(\boldsymbol{\theta}^{t+1})\right] - \mathcal{L}^o(\boldsymbol{\theta}^t) \leq -\frac{\eta}{2}\left\|\nabla\mathcal{L}^o(\boldsymbol{\theta}^t)\right\|^2 + \frac{\eta^2 L}{2}\left(\sigma_b^2 + K\cdot\sigma_s^2\right).$$

Taking the unconditional expectation (over $\mathcal{K}_o$ and $\mathcal{F}^{t+1}$) on both sides of this inequality, we get:

$$\mathbb{E}\left[\mathcal{L}^o(\boldsymbol{\theta}^{t+1}) - \mathcal{L}^o(\boldsymbol{\theta}^t)\right] \leq -\frac{\eta}{2}\cdot\mathbb{E}\left[\left\|\nabla\mathcal{L}^o(\boldsymbol{\theta}^t)\right\|^2\right] + \frac{\eta^2 L}{2}\left(\sigma_b^2 + K\cdot\sigma_s^2\right).$$

Now, using the law of total expectation, we get that

$$\mathbb{E}\left[\mathbb{E}\left[\mathcal{L}^o(\boldsymbol{\theta}^{t+1}) - \mathcal{L}^o(\boldsymbol{\theta}^t)\mid\mathcal{F}^{t+1}\right]\right] \leq -\frac{\eta}{2}\cdot\mathbb{E}\left[\mathbb{E}\left[\left\|\nabla\mathcal{L}^o(\boldsymbol{\theta}^t)\right\|^2\mid\mathcal{F}^{t+1}\right]\right] + \frac{\eta^2 L}{2}\left(\sigma_b^2 + K\cdot\sigma_s^2\right).$$

Therefore, using the fact that $\mathcal{L}(\boldsymbol{\theta}) = \mathbb{E}[\mathcal{L}^o(\boldsymbol{\theta})]$ and Jensen's inequality (12), we have

$$\mathbb{E}\mathcal{L}(\boldsymbol{\theta}^{t+1}) - \mathbb{E}\mathcal{L}(\boldsymbol{\theta}^t) \le -\frac{\eta}{2} \cdot \mathbb{E}\left\|\mathbb{E}\left[\nabla\mathcal{L}^o(\boldsymbol{\theta}^t) \mid \mathcal{F}^{t+1}\right]\right\|^2 + \frac{\eta^2 L}{2}\left(\sigma_b^2 + K \cdot \sigma_s^2\right).$$

Using the dominated convergence theorem, we can interchange the expectation and the gradient operators, arriving at the following descent lemma in expectation:

$$\mathbb{E}\mathcal{L}(\boldsymbol{\theta}^{t+1}) - \mathbb{E}\mathcal{L}(\boldsymbol{\theta}^t) \le -\frac{\eta}{2} \cdot \mathbb{E}\left\|\nabla\mathcal{L}(\boldsymbol{\theta}^t)\right\|^2 + \frac{\eta^2 L}{2}\left(\sigma_b^2 + K \cdot \sigma_s^2\right). \tag{17}$$

Taking the average of the inequality above for $t \in \{0, 1, \ldots, T-1\}$, we get that:

$$\frac{\mathbb{E}\mathcal{L}(\boldsymbol{\theta}^T) - \mathcal{L}(\boldsymbol{\theta}^0)}{T} \le -\frac{\eta}{2T}\sum_{t=0}^{T-1}\mathbb{E}\left\|\nabla\mathcal{L}(\boldsymbol{\theta}^t)\right\|^2 + \frac{\eta^2 L}{2}\left(\sigma_b^2 + K \cdot \sigma_s^2\right).$$

Lastly, rearranging the terms and using the existence and definition of $\mathcal{L}^\star$ (A1), we have that

$$\frac{1}{T}\sum_{t=0}^{T-1}\mathbb{E}\left\|\nabla\mathcal{L}(\boldsymbol{\theta}^t)\right\|^2 \le \frac{2(\mathcal{L}(\boldsymbol{\theta}^0) - \mathcal{L}^\star)}{\eta T} + \eta L\left(\sigma_b^2 + K \cdot \sigma_s^2\right),$$

thus arriving at the result in (10).

## C.4    PROOF OF THEOREM 2

Following the same steps as in the proof of Theorem 1, we can arrive at the same descent lemma in expectation (17),

$$\mathbb{E}\mathcal{L}(\boldsymbol{\theta}^{t+1}) - \mathbb{E}\mathcal{L}(\boldsymbol{\theta}^t) \le -\frac{\eta}{2} \cdot \mathbb{E}\left\|\nabla\mathcal{L}(\boldsymbol{\theta}^t)\right\|^2 + \frac{\eta^2 L}{2}\left(\sigma_b^2 + K \cdot \sigma_s^2\right).$$

Rearranging the terms and subtracting the infimum on both sides, we get that

$$\mathbb{E}\mathcal{L}(\boldsymbol{\theta}^{t+1}) - \mathcal{L}^\star \le \mathbb{E}\mathcal{L}(\boldsymbol{\theta}^t) - \mathcal{L}^\star - \frac{\eta}{2} \cdot \mathbb{E}\left\|\nabla\mathcal{L}(\boldsymbol{\theta}^t)\right\|^2 + \frac{\eta^2 L}{2}\left(\sigma_b^2 + K \cdot \sigma_s^2\right).$$

Now, using the definition of our Lyapunov function, $\delta^t = \mathbb{E}\mathcal{L}(\boldsymbol{\theta}^t) - \mathcal{L}^\star$, we arrive at the following inequality:

$$\delta^{t+1} \le \delta^t - \frac{\eta}{2} \cdot \mathbb{E}\left\|\nabla\mathcal{L}(\boldsymbol{\theta}^t)\right\|^2 + \frac{\eta^2 L}{2}\left(\sigma_b^2 + K \cdot \sigma_s^2\right).$$

Now, from the PL inequality (A4), we have that:

$$\delta^{t+1} \le \delta^t - \mu\eta \cdot \mathbb{E}\left[\mathcal{L}(\boldsymbol{\theta}^t) - \mathcal{L}^\star\right] + \frac{\eta^2 L}{2}\left(\sigma_b^2 + K \cdot \sigma_s^2\right).$$

Therefore, again using the definition of our Lyapunov function, we arrive at

$$\delta^{t+1} \le (1 - \mu\eta) \cdot \delta^t + \frac{\eta^2 L}{2}\left(\sigma_b^2 + K \cdot \sigma_s^2\right).$$

Recursing the inequality above, we get that

$$\delta^T \le (1 - \mu\eta)^T \delta^0 + \frac{\eta^2 L}{2}\left(\sigma_b^2 + K \cdot \sigma_s^2\right)\sum_{t=0}^{T-1}(1 - \mu\eta)^t.$$

Finally, using the sum of a geometric series, we arrive at

$$\delta^T \le (1 - \mu\eta)^T \delta^0 + \frac{\eta L}{2\mu}\left(\sigma_b^2 + K \cdot \sigma_s^2\right),$$

which corresponds to the result that we set out to prove.

## D EXPERIMENT DETAILS

**Notes on `VFedTrans`.** For the `VFedTrans` (Huang et al., 2023) pipeline, we use `FedSVD` (Chai et al., 2022) for federated representation learning and an autoencoder for local representation learning, as these are the best performing methods in Huang et al. (2023). For the local predictor, we use a multilayer perceptron, as in Huang et al. (2023). As seen in Table 1, the performance of `VFedTrans` is nearly identical to that of `Local`. However, given that requires a multi-stage pipeline with operations for each pair of clients, its runtime is much longer. In particular, for the Credit dataset, for $p_{\text{miss}} = 0.0$ for training and inference, running `VFedTrans` takes $2843.0 \pm 50.6$s, while running `Local` takes only $127.8 \pm 3.0$s.

**Notes on `PlugVFL`.** We also note that the original `PlugVFL` paper (Sun et al., 2024) **1)** does not address missing training data and **2)** only considers settings with two clients. In our experiments, we extended `PlugVFL` to handle more generic scenarios. To address missing training data fairly, we replaced representations for missing feature blocks with zeros at the fusion model, similarly to the dropout mechanisms in the original paper, but now, for missing data, these representations are zeroed throughout training rather than per iteration. To generalize `PlugVFL` for more than two clients, we assigned a dropout probability to each passive party (clients not holding the fusion model).

**Notes on MIMIC-IV.** We use ICU data of MIMIC-IV v1.0 (Johnson et al., 2020) and follow the data processing pipeline in Gupta et al. (2022). We focus on the task of predicting in-hospital mortality for patients admitted with chronic kidney disease. We do feature selection with diagnosis data and the selected features of chart events (labs and vitals) used in the "Proof of concept experiments" Gupta et al. (2022) as input features.

**Computing metrics.** As mentioned in the main text of the paper, we want our metrics to capture that fact that we want all clients to perform well at inference time. Therefore we consider the average metrics across the clients. More precisely, we compute the metrics for Table 1 as follows:

- For the test accuracy, let $\hat{y}^n(k)$ denote the prediction of client $k$ for sample $n$, we compute:

$$\text{accuracy} = 100 \times \frac{1}{N} \sum_{n=1}^{N} \left( \frac{1}{|\mathcal{K}_o^n|} \sum_{k \in \mathcal{K}_o^n} \mathbf{1}(\hat{y}^n(k) = y^n) \right).$$

- For the $F_1$-score, we compute:

$$F_1\text{-score} = \frac{1}{K} \sum_{k=1}^{K} \frac{P_k \cdot R_k}{P_k + R_k},$$

where $P_k$ and $R_k$ are the precision and recall of client $k$, with respect to its predictor and (only) the samples it observed.

**Models used.** For the HAPT and Credit datasets, we trained simple multilayer perceptrons; for the MIMIC-IV dataset, we trained an LSTM (Hochreiter, 1997); and, for the CIFAR-10 and CIFAR-100 experiments, we use ResNets18 (He et al., 2016) models.

## E ADDITIONAL EXPERIMENTS

In this appendix, we present additional experimental results.

**Nonuniform missing features.** In Table 3, we present an experiment where the probability of each feature block $k$ not being observed, $p_{\text{miss}}(k)$, is sampled independent and identically distributed following $p_{\text{miss}}(k) \sim \text{Beta}(2.0, 2.0)$. Note that, for $X \sim \text{Beta}(2.0, 2.0)$, we have that $\mathbb{E}[X] = 0.5$. This motivates our comparison with $p_{\text{miss}}(k) = 0.5$ for all $k$. For the column corresponding to the case where both the probability of missing training data and inference data are nonuniform, these probabilities are sampled independently for training and inference.

As in the previous experiments, we see that LASER-VFL outperforms the baselines. Further, not only does the nonuniform nature of $p_{\text{miss}}(k)$ not harm performance, but in fact our LASER-VFL method seems to do slightly better when the feature blocks have missing training data with a probability $p_{\text{miss}}(k) \sim \text{Beta}(2.0, 2.0)$ rather than when we simply have $p_{\text{miss}}(k) = 0.5$, which corresponds to the expected value of the random variable $p_{\text{miss}}(k)$.

Table 3: Test accuracy for different probabilities of missing blocks during training and inference (including nonuniform) for CIFAR-10 with $K = 4$ clients, averaged over five seeds ($\pm$ standard deviation).

| Training $p_{\text{miss}}(k)$ | $\sim \text{Beta}(2.0,2.0)$ | | | | 0.5 | | |
|---|---|---|---|---|---|---|---|
| Inference $p_{\text{miss}}(k)$ | 0.0 | 0.1 | 0.5 | $\sim \text{Beta}(2.0,2.0)$ | 0.0 | 0.1 | 0.5 |
| CIFAR-10 (accuracy, %) | | | | | | | |
| Local | $72.7 \pm 1.9$ | $72.8 \pm 1.8$ | $72.6 \pm 2.2$ | $73.4 \pm 1.0$ | $71.2 \pm 0.4$ | $71.1 \pm 0.4$ | $71.3 \pm 0.7$ |
| Standard VFL | $68.2 \pm 11.6$ | $54.0 \pm 15.7$ | $13.9 \pm 5.4$ | $13.5 \pm 5.3$ | $54.9 \pm 10.0$ | $38.6 \pm 9.1$ | $10.9 \pm 2.2$ |
| Ensemble | $83.9 \pm 1.7$ | $82.7 \pm 2.0$ | $75.3 \pm 2.6$ | $75.4 \pm 1.7$ | $82.4 \pm 0.2$ | $81.0 \pm 0.8$ | $73.8 \pm 0.6$ |
| Combinatorial | $67.8 \pm 11.5$ | $70.2 \pm 10.1$ | $73.4 \pm 6.6$ | $75.9 \pm 2.9$ | $54.9 \pm 9.7$ | $58.6 \pm 7.7$ | $68.4 \pm 3.1$ |
| PlugVFL | $80.9 \pm 2.1$ | $80.1 \pm 1.7$ | $77.8 \pm 0.4$ | $77.1 \pm 0.1$ | $76.5 \pm 2.1$ | $75.6 \pm 1.5$ | $72.4 \pm 1.0$ |
| LASER-VFL (ours) | $\mathbf{88.8} \pm 1.4$ | $\mathbf{88.2} \pm 1.6$ | $\mathbf{81.1} \pm 2.3$ | $\mathbf{81.7} \pm 1.0$ | $\mathbf{87.4} \pm 0.3$ | $\mathbf{86.4} \pm 0.6$ | $\mathbf{79.4} \pm 1.6$ |

