# OpenReview forum: "Vertical Federated Learning with Missing Features During Training and Inference"
_ICLR.cc/2025/Conference — ICLR 2025 Poster_

### Official Review · Reviewer_Cmzg · 2024-11-01

**Soundness:** 2
**Presentation:** 3
**Contribution:** 3
**Rating:** 6
**Confidence:** 4

**Summary:**

This paper focuses on vertical federated learning. More specifically, the authors present a framework called LASER-VFL, which is more robust in handling missing features and any arbitrary combination of partitions. They provide a convergence rate analysis along with  experimental results.

**Strengths:**

I really enjoyed reading this paper. The presentation is good and easy to follow, and the experimental results are very convincing. More specifically:
- The paper addresses missing features problem, which is an important area of study in VFL.
- The proposed idea is simple yet effective. The authors create a fusion of arbitrary combinations of predictors on each client, allowing them to leverage all available features. To avoid sharing an exponential number of predictors, they propose a strategy in which only K predictors and K fusion models need to be shared.
- They demonstrate improved performance compared to the existing works.

**Weaknesses:**

- While the authors present related works, it is difficult to identify which components of the proposed framework are novel when reading Sections 2 and 3. I encourage the authors to provide a more detailed comparison with existing VFL approaches that address missing features.
- The results are convincing, especially given that they cover different levels of missing features. However, only two VFL methods are used for comparison. The authors are encouraged to include additional comparisons to better highlight the work's significance in the field of VFL.

**Questions:**

- Why are there K fusion models?  Is it because each client needs a specialized fusion model for its own task?
- Could the authors elaborate on line 2 in Algorithm 1 (Client K sample the same mini-batch .... via a shared seed)? What is the impact of this shared seed, and why is the same mini-batch important?
- Could the authors elaborate on how domain shift is addressed? Or how would the framework generalize if there were a significant level of domain shift across clients?
- Does the proposed framework have specific requirements for the fusion model and projectors? Also, what is the architecture of the fusion model and predictors in Section 5?

---

> ### Author Response · Authors · 2024-11-22
>
> Thank you for your comments.
>
> ### Re: a more detailed comparison with existing approaches
> The reason why we did not perform a more detailed comparison with individual existing VFL methods addressing either missing features during training or during inference is that these differ very significantly from our method (and often from each other); they have little more in common with our work than their objective. In fact, even in terms of the task being tackled, we believe our method is the first one addressing _both_ of these challenges simultaneously (missing features during training _and_ during inference). To the best of our knowledge, the whole idea of modeling predictions from different subsets of features as separate tasks, each with its own loss term, and then sampling from these tasks to make training feasible, is novel.
>
> ### Re: additional baselines
>
> We have added a new, strong baseline to Table 1: the combinatorial approach described in Section 2. These new results show that our method outperforms even the combinatorial approach. We attributed this to (1) the presence of missing features during training, meaning that in the combinatorial approach, for each batch, we can only train the predictors corresponding to a set of features that is a subset of the set of observed features, significantly decreasing the number of training samples and thus achieving worse generalization; and (2) even in the absence of missing training data, our LASER-VFL method still outperforms the combinatorial approach due to the dropout-like regularization effect introduced by task sampling.
>
> ### Re: why $K$ fusion models
>
> The main reason for having $K$ fusion models, one at each client, is to ensure that any client can continue performing the prediction task it was trained for during collaborative VFL (whether or not it is the same task as other clients) regardless of whether any other client leaves the federation. If a single fusion model were trained at a given client and that client were to leave the federation during inference, this would render the models of all the other clients unavailable during inference. Therefore, having a local fusion model gives more insurance of continued model availability during inference to each of the clients. Nevertheless, as you mentioned, this also allows each client to have a specialized fusion model for its own task, which can be very convenient.
>
> ### Re: on the use of a shared seed
>
> The use of a shared seed allows the different clients to randomly sample the same indices for each mini-batch without having to communicate the batch indices at each communication round; this is common in VFL [1, 2]. This matching mini-batch is important because otherwise we would train the model on invalid samples. To illustrate this, consider a dataset with only two samples
> $\lbrace \mathbf{x}_1, \mathbf{x}_2 \rbrace$ with each sample $\mathbf{x}_i$ further partitioned into two blocks of features $\mathbf{x}\_i=(\mathbf{x}\_{i1}, \mathbf{x}\_{i2})$. If a client holding $\lbrace\mathbf{x}\_{i1}:\forall i\rbrace$ samples a sample (mini-batch of size one) without ensuring it matches the sample of the another client holding $\lbrace\mathbf{x}\_{i2}:\forall i\rbrace$, we may end up computing the loss corresponding to a prediction based on $(\mathbf{x}\_{i1}, \mathbf{x}\_{j2})$, where $i\not=j$, which is not a valid sample.
>
> ### Re: domain shift
>
> As mentioned above, while having a fusion model at each client allows each client to tackle a different task, this is not the focus of this work. For this reason, we did not explore such a domain shift across clients, leaving it as future work instead.
>
> ### Re: requirements on the model architecture
>
> Our framework is very flexible regarding the types of models it can accommodate. In the experiments, we used a variety of representation models, including multilayer perceptrons, an LSTM, and ResNets. For the fusion models, we employed a simple linear transformation (a single linear layer neural network), as it was sufficient to achieve good performance; however, other fusion models can also be considered, as our LASER-VFL method does _not_ restrict the fusion model to consist of a single layer. We have added more details about our experiments in the supplementary material of the revised manuscript.
>
> ### References
>
> [1] Castiglia, Timothy, Shiqiang Wang, and Stacy Patterson. "Flexible vertical federated learning with heterogeneous parties." IEEE Transactions on Neural Networks and Learning Systems (2023).
>
> [2] Valdeira, Pedro, et al. "Communication-efficient Vertical Federated Learning via Compressed Error Feedback." arXiv preprint arXiv:2406.14420 (2024).

---

> > ### Comment · Reviewer_Cmzg · 2024-11-24
> >
> > I thank the author for their response. While I understand that the scope and methodology of prior VFL approaches that address missing features may differ from yours, not comparing against them as having “little more in common with your work than their objective” undermines the contributions of these works.
> >
> > Additionally, the authors mention many times that LASER-VFL is the first to address both challenges (missing features during training and inference) simultaneously. This claim would be stronger if supported by a concrete comparison with the existing VFL methods even though those methods only address one of the aspects.
> >
> > Therefore, I will keep my score unchanged.

---

> > > ### Author Response · Authors · 2024-11-24
> > >
> > > Many thanks for your reply and further suggestions.
> > >
> > > ### Re: further baselines
> > >
> > > We have added a new baseline to the experiments in Section 5: PlugVFL [3]. PlugVFL is the most highly cited of the three works enabling inference in VFL from varying sets of feature blocks. After recently obtaining the code from the authors, we were able to include PlugVFL as a baseline in our experiments. While PlugVFL outperforms most baselines, it does so inconsistently and our LASER-VFL method ultimately outperforms PlugVFL, as shown in the revised Table 1 and in the experiments with a varying number of clients.
> > >
> > > Note that we have made the necessary extensions to PlugVFL to support missing training data and more than two clients, which were not considered in the approach described in the original paper. See Appendix D of the revised manuscript for details.
> > >
> > > We would also like to explain that the reason for not including more baselines in our original submission was the difficulty of reproducing results from the related methods, due to the unavailability of their source code. In this spirit, we are happy to share our code and make it open-source upon acceptance, to enable others to compare their methods with ours. In the meantime, we have made our code publicly available in an anonymos GitHub repository:
> > > https://anonymous.4open.science/r/robust-vfl-7A1E/README.md
> > >
> > > ### References
> > >
> > > [3] Jingwei Sun, Zhixu Du, Anna Dai, Saleh Baghersalimi, Alireza Amirshahi, Qilin Zheng, David Atienza, & Yiran Chen. (2024). PlugVFL: Robust and IP-Protecting Vertical Federated Learning against Unexpected Quitting of Parties.

---

> > > > ### Comment · Reviewer_Cmzg · 2024-11-29
> > > >
> > > > Thank you for providing additional clarification regarding the lack of comparison with existing work. However, I notice that your current reasoning—that it is difficult to replicate prior methods—differs from your initial response, which emphasized differences in scope and methodology. This inconsistency is concerning, as it suggests that the novelty and positioning of your work may not have been fully established in relation to existing approaches. Therefore, I suggest that the authors rigorously address this issue in future revisions of the manuscript.

---

> > > > > ### Author Response · Authors · 2024-11-29
> > > > >
> > > > > Thank you again for engaging in this discussion. We do not find the two reasonings you mentioned to be inconsistent. Instead, they are two reasons that collectively explain why we did not add the additional baselines initially. We discuss this further in the following.
> > > > >
> > > > > The difference in scope between our work and prior art, where the prior art addressed missing features in VFL during either training or inference but does not tackle both challenges simultaneously, explains why we initially deemed a less exhaustive comparison with prior works acceptable. This decision was based on the reasonable expectation that our method would outperform existing approaches for the specific task it was designed to address—missing features during both training and inference. This conclusion has since been validated by our more extensive experimental results. Nonetheless, had it not been for the second issue we raised—the challenge of replicating these baselines due to the lack of publicly available code—we would have included more baselines from the start. We believe that such comparisons strengthen the contribution of the paper, even if these baselines are not fully in scope. Allow us to elaborate further.
> > > > >
> > > > > On the one hand, methods designed to handle only missing training data result in a joint predictor that becomes unavailable for inference when features are missing during inference. This limitation is evident in the sharp performance deterioration observed in `Standard VFL` when there is no missing training data but there are missing features during inference.
> > > > >
> > > > > On the other hand, most methods dealing (only) with missing test data result in local predictors that cannot leverage features observed by other clients during inference. We confirmed the negative impact of this empirically in the original submission, as we included `VFedTrans` [4] as a baseline. As expected, `VFedTrans` slightly outperforms the `Local` approach, but it wastes a lot of information when more feature blocks are observed during inference, leading to poor performance compared to methods that do not waste these additional feature blocks. Namely, the `Ensemble` baseline outperforms this method significantly, as it leverages information from all the clients during inference. For this reason, we expect the `Ensemble` method to also outperform other methods that train local predictors.
> > > > >
> > > > > The notable exception to methods addressing missing features during inference by training local predictors is the small group of recent works that handle a varying number of feature blocks at inference time in VFL [3, 5, 6], as mentioned in our previous message. However, as discussed in the previous paragraph, these methods do not address missing training data and are therefore also expected to be outperformed by our `LASER-VFL` method in scenarios involving such missing data.
> > > > >
> > > > > Nevertheless, _we acknowledge that a more detailed comparison with these methods was lacking in the original submission_. This has been addressed in the revised version of the manuscript. Therefore, this is not a weakness in the current version of the paper. In particular, as explained in our previous comment:
> > > > > >We have added `PlugVFL` [3] as a baseline, and have observed that our `LASER-VFL` method outperforms it. We also note that the paper by Ganguli et al. [5] is equivalent to `PlugVFL` in our setting (i.e., the case where all clients can always communicate with each other, which is prevalent in VFL). Finally, as we now explain in our revised manuscript (in Section 1), the work in [6] only considers binary classification and, since their algorithm relies on results from [7] which explicitly requires the assumption of binary classification, it is not straightforward to extend their method to non-binary tasks.
> > > > >
> > > > > Therefore, we do not believe there is any inconsistency in our answers (though we acknowledge that we could have elaborated further, for clarity). Moreover, given that our current comparison with prior works is comprehensive, we believe the positioning of our work is now well established in relation to these existing approaches.
> > > > >
> > > > > ### References
> > > > >
> > > > > [3] Jingwei Sun, et al. (2024). PlugVFL: Robust and IP-Protecting Vertical Federated Learning against Unexpected Quitting of Parties.
> > > > >
> > > > > [4] Huang, Chung-ju, et al. "Vertical federated knowledge transfer via representation distillation for healthcare collaboration networks." Proceedings of the ACM Web Conference 2023. 2023.
> > > > >
> > > > > [5] Ganguli, Surojit, et al. "Fault-tolerant vertical federated learning on dynamic networks." arXiv preprint arXiv:2312.16638 (2023).
> > > > >
> > > > > [6] Gao, Dashan, et al. "Complementary Knowledge Distillation for Robust and Privacy-Preserving Model Serving in Vertical Federated Learning." Proceedings of the AAAI Conference on Artificial Intelligence. Vol. 38. No. 18. 2024.
> > > > >
> > > > > [7] Zhang, Harry. "The Optimality of Naive Bayes." Proceedings of the the 17th International FLAIRS conference (FLAIRS2004). 2004.

---

> > > ### Author Response · Authors · 2024-11-27
> > >
> > > Thank you again for engaging in the discussion!
> > >
> > > Upon further reflection on our experiments, we would like to provide an additional clarification regarding the completeness of the baselines included in the revised version of the paper. The main updates to the revised manuscript are highlighted in blue for your convenience.
> > >
> > > ### Re: completeness of baselines included in the experiments
> > >
> > > On top of comparing our method to the `Local` approach and to `Stardard VFL`, which are standard baselines for work in this area, we have included a reasonable and effective baseline consisting of an `Ensemble` method.
> > >
> > > We then move to works that specifically target inference in VFL in the presence of missing features. First, we include `VFedTrans` [4], which is a recent method from the line of work that leverages information from the entire federation to train a local predictor (i.e., a predictor that depends only on the local features of its client during inference). We observe that, as expected, while such methods can slightly outperform the `Local` approach, they waste a lot of information when more feature blocks are observed during inference. For this reason, we see that the `Ensemble` baseline outperforms this method, as it leverages information from all the clients during inference.
> > >
> > > Afterwards, we compare with methods allowing for a varying number of feature blocks during inference. We start by including the `Combinatorial` baseline, which explicitly trains a predictor for each possible combination of missing features. We observe that, in addition to having much better scalability, our `LASER-VFL` method outperforms `Combinatorial`. Then, we look at the existing methods supporting a varying number of available feature blocks in the literature. To the best of our knowledge, there are only three papers dealing with a varying number of features blocks at inference time in VFL [3, 5, 6]. We have added `PlugVFL` [3] as a baseline, and have observed that our `LASER-VFL` method outperforms it. We also note that the paper by Ganguli et al. [5] is equivalent to `PlugVFL` in our setting (i.e., the case where all clients can always communicate with each other, which is prevalent in VFL). Finally, as we now explain in our revised manuscript (in Section 1), the work in [6] only considers binary classification and, since their algorithm relies on results from [7] which explicitly requires the assumption of binary classification, it is not straightforward to extend their method to non-binary tasks.
> > >
> > > Therefore, we believe that the _current baselines included in our experiments in Section 5 should be considered as complete_. These results allow us to claim that our method outperforms state-of-the-art methods for related problems in the presence of missing features. This is a significant and timely contribution to VFL, which we believe makes our work worthy of a (more than borderline) acceptance.
> > >
> > > Please let us know if you have any further comment or question.
> > >
> > > ### References
> > >
> > > [3] Jingwei Sun, Zhixu Du, Anna Dai, Saleh Baghersalimi, Alireza Amirshahi, Qilin Zheng, David Atienza, & Yiran Chen. (2024). PlugVFL: Robust and IP-Protecting Vertical Federated Learning against Unexpected Quitting of Parties.
> > >
> > > [4] Huang, Chung-ju, Leye Wang, and Xiao Han. "Vertical federated knowledge transfer via representation distillation for healthcare collaboration networks." Proceedings of the ACM Web Conference 2023. 2023.
> > >
> > > [5] Ganguli, Surojit, et al. "Fault-tolerant vertical federated learning on dynamic networks." arXiv preprint arXiv:2312.16638 (2023).
> > >
> > > [6] Gao, Dashan, et al. "Complementary Knowledge Distillation for Robust and Privacy-Preserving Model Serving in Vertical Federated Learning." Proceedings of the AAAI Conference on Artificial Intelligence. Vol. 38. No. 18. 2024.
> > >
> > > [7] Zhang, Harry. "The Optimality of Naive Bayes." Proceedings of the the 17th International FLAIRS conference (FLAIRS2004). 2004.

---

> ### Author Response · Authors · 2024-12-01
> **Discussion Reminder**
>
> Dear Reviewer Cmzg,
>
> Thank you again for your time and effort in reviewing our paper. Since tomorrow is the final day for reviewers to post comments, we would appreciate your feedback on whether our responses to your comments and revisions to the manuscript are satisfactory. If you have any further questions, please let us know and we will be happy to provide additional clarifications. If not, we would be grateful if you could consider increasing your score to reflect the fact that we have addressed your concerns.
>
> Best regards,
> The Authors

---

### Official Review · Reviewer_GeBH · 2024-11-02

**Soundness:** 3
**Presentation:** 2
**Contribution:** 2
**Rating:** 6
**Confidence:** 3

**Summary:**

The author(s) proposed LASER-VFL to handle the missing feature challenge arising from the vertical federated learning problem. By transforming the objective into a finite sum problem, LASER-VFL can handle missing features (clients) via the task sampling technique, which is similar to client dropout. LASER-VFL is backed with theoretical analysis and some simple numerical experiments.

**Strengths:**

- The writing is clear in general;
- The paper is well-motivated, the missing feature during training and inference is a crucial problem in vertical federated learning from a practical perspective.
- The proposed method is simple and easy to understand.

**Weaknesses:**

- The experiment section is rather weak:
    - Experiments details are missing, for example, the train/test split of dataset, the neural network architecture being used, the hyper-parameter tunning...
    - A baseline is missing, the naïve method described in section 2 (train exponentially many predictors).
    - The number of clients is set to 4 for all experiments, how does the result change when we increase the number of clients? If there are 4 clients, then it is not really computational prohibitive to train exponentially many predictors.
- Notations are quite heavy in line 293-342.
- The theoretical analysis assumes features are missed uniform randomly, which simplifies the problem significantly. By this assumption and transforming the objective into a finite sum problem, the author(s) could directly use classic convergence analysis of SGD without much technical modification. Therefore, the theoretical novelty is kind of limited.
- Minor: missing reference at line 855.

**Questions:**

By transforming the objective into a finite sum problem, is the convergence analysis almost the same as the classic convergence analysis of SGD?

---

> ### Author Response · Authors · 2024-11-22
>
> Thank you for your comments.
>
> ### Re: weaknesses in the experiments section
>
> We have significantly strengthened the experiments section. In particular, we have made the following  improvements:
> * We have added some details on the experiments in the supplementary material (given the space contraints) and we will be making our code publicly available upon acceptance, thus ensuring reproducibility.
> * We have added the naive combinatorial method described in Section 2 to Table 1 and we have added new experiments comparing this combinatorial approach to our method and to the other baselines for a range of numbers of clients (see Section 5 of the revised manuscript). These experiments show that, even for a small number of clients, the combinatorial approach is significantly slower than our method, as seen in the following table, where we present the runtime of the combinatorial approach divided by the runtime of our LASER-VFL method for CIFAR-10:
>
>     | Number of Clients (K) | 2    | 3    | 4    | 5     | 6     | 7     |
>     |------------------------|------|------|------|-------|-------|-------|
>     | Runtime Ratio          | 1.76 | 3.19 | 5.96 | 10.91 | 19.41 | 35.80 |
>
>     This demonstrates that the combinatorial approach may become impractical even for a modest number of clients. Moreover, the combinatorial approach is not optimal in terms of performance: (1) in the presence of missing features during training, for each batch, it can only train the predictors using a set of features that is a subset of the set of observed features, significantly decreasing the number of training samples and thus achieving worse generalization; and (2) even when there is no missing training data, our LASER-VFL method still outperforms the combinatorial approach due to the dropout-like regularization effect introduced by task sampling.
> * We have added a new baseline to the experiments in Section 5: PlugVFL [1]. PlugVFL is the most highly cited of the three works enabling inference in VFL from varying sets of feature blocks. After recently obtaining the code from the authors, we were able to include PlugVFL as a baseline in our experiments. While PlugVFL outperforms most baselines, it does so inconsistently and our LASER-VFL method ultimately outperforms PlugVFL, as shown in the revised Table 1 and in the experiments with a varying number of clients.
>
>     Note that we have made the necessary extensions to PlugVFL to support missing training data and more than two clients, which were not considered in the approach described in the original paper. See Appendix D of the revised manuscript for details.
>
>     We would also like to explain that the reason for not including more baselines in our original submission was the difficulty of reproducing results from the related methods, due to the unavailability of their source code. In this spirit, we are happy to share our code and make it open-source upon acceptance, to enable others to compare their methods with ours. In the meantime, we have made our code publicly available in an anonymos GitHub repository:
>     https://anonymous.4open.science/r/robust-vfl-7A1E/README.md
>
> ### Re: heavy notation
>
> We have worked to simplify the notation in the revised manuscript and improve the readability of the section you mentioned, namely by reducing the use of notation. (See the "Our optimization method" paragraph in Section 3.2). However, given the topic of the paper, further simplification is challenging, as the use of indices for sets of models, feature blocks, and stochastic approximations is often essential for clarity and precision.
>
> ### Re: theoretical analysis
>
> Our key theoretical contribution lies in framing the problem of missing features in vertical federated learning as a finite-sum problem. This formulation allows us to leverage analytical tools developed for finite-sum problems to establish convergence guarantees for our method, thus confirming its validity and supporting its design under uniformly missing blocks of features—a desirable property. This stands in contrast to previous works on vertical federated learning enabling a varying number of clients to collaborate during inference, which do not provide convergence guarantees [1, 2].
>
> ### Re: missing reference
>
> Thank you for bringing the missing reference to our attention.
>
> ### References
>
> [1] Jingwei Sun, Zhixu Du, Anna Dai, Saleh Baghersalimi, Alireza Amirshahi, Qilin Zheng, David Atienza, & Yiran Chen. (2024). PlugVFL: Robust and IP-Protecting Vertical Federated Learning against Unexpected Quitting of Parties.
>
> [2] Gao, Dashan, et al. "Complementary Knowledge Distillation for Robust and Privacy-Preserving Model Serving in Vertical Federated Learning." Proceedings of the AAAI Conference on Artificial Intelligence. Vol. 38. No. 18. 2024.

---

> > ### Comment · Reviewer_GeBH · 2024-12-02
> > **Thanks for the revision**
> >
> > I would like to thank the author(s) for improving the experiments section. I am raising my score.

---

> > > ### Author Response · Authors · 2024-12-02
> > > **Thanks for the response**
> > >
> > > Thank you for your response and for raising the score; we appreciate it.

---

> ### Author Response · Authors · 2024-11-28
> **Discussion Reminder**
>
> Dear Reviewer GeBH,
>
> Thank you again for your valuable comments and feedback. We hope that our response has addressed your concerns and clarified your questions. As the discussion period comes to a close, please feel free to reach out if you have any additional questions.
>
> Best regards,
> The Authors

---

> ### Author Response · Authors · 2024-12-01
> **Discussion Reminder**
>
> Dear Reviewer GeBH,
>
> Thank you again for your time and effort in reviewing our paper. Since tomorrow is the final day for reviewers to post comments, we would appreciate your feedback on whether our responses to your comments and revisions to the manuscript are satisfactory. If you have any further questions, please let us know and we will be happy to provide additional clarifications. If not, we would be grateful if you could consider increasing your score to reflect the fact that we have addressed your concerns.
>
> Best regards,
> The Authors

---

### Official Review · Reviewer_wVzL · 2024-11-02

**Soundness:** 3
**Presentation:** 3
**Contribution:** 2
**Rating:** 6
**Confidence:** 4

**Summary:**

This paper proposes to handle missing features in VFL during both training and testing by sharing model parameters and sampling different configurations to approximate the exponential number of possible predictors (with different possible non-missing features).
The intuition is that entirely local models are robust to missing features but poor predictors while global models are stronger predictors but not robust missing features.
The paper proposes to train a set of models for all possible subsets of missing features by sharing model parameters and sampling different missingness patterns, called "tasks" in this paper, during training time.
The sampled tasks are weighted to approximate the loss function of all missing value patterns using Monte Carlo sampling of the tasks.
The paper provides theoretic results on convergence and empirically demonstrates their method compared to several baseline methods.

**Strengths:**

- The paper proposes a Monte-Carlo approach with weighting to train a split-NN architecture to be robust to missing features during training and inference. The method is relatively simple and avoids complex hyperparameters or multi-stage configurations.

- The paper provides theoretic convergence results on their proposed method.

- The empirical results show significant improvement over baselines methods used in the paper when there is missingness at inference time.

**Weaknesses:**

------ After reading author response ------
While I still have some concerns, the author response helped reduce them. Thus, I've increased my score by 1.
------ Original -------
- Either the setting is a small number of clients (say 4 as used in the experiments) and then it might be feasible to actually just train all possible models---this calls into question the motivation for learning "an exponential number of models" since there are only 24 models. Or, the setting is a large number of clients (say 10-100) and then finding large batches of samples such that the missing values are the same would result in either small or biased batches---which would violate assumption 2. Thus, the motivation of the paper or the requirement for large unbiased batches seem to be at odds with each other.

- The assumption of unbiased batches would require the missingness to be Missing Completely at Random (MCAR) to ensure that the batches are unbiased. This seems unlikely to hold in real-world applications.

- The paper seems to overclaim a bit on the novelty or significance of the model sharing aspect. The model sharing aspect is a common approach in GNNs or other modern networks such as attention-based architectures, yet these do not claim significant novelty in terms of sharing model parameters across nodes.

- The paper lacks an empirical comparison to the method proposed in Ganguli et al. [2023] that can handle inference time faults. This seems to be a closely related method. They use dropout (like your task sampling), gossiping (like the ensemble-based baseline), and replication of the fusion/predictor (as you do as well).

[Ganguli et al., 2023] Ganguli, S., Zhou, Z., Brinton, C. G., & Inouye, D. I. (2023). Fault Tolerant Serverless VFL Over Dynamic Device Environment. arXiv preprint arXiv:2312.16638.

**Questions:**

- How does this paper compare to the methods proposed in Ganguli et al., [2023]? They seem to be doing something very similar including the dropout mechanism during training. Perhaps, the authors missed this very related work.

- The model architecture resembles a single-layer GNN with summation aggregation. Each client has it's own fusion predictor. Is this correct or am I missing something?

- While the weighting method for learning an "exponential number of models" is interesting, it is perhaps a bit misleading to argue for the "compactness" of learning an exponential number of models. Simple dropout could also learn "an exponential number of models" but it is difficult to argue that these models would be anywhere near the naive solution of actually directly training an exponential number of models.

- There are other types of parametric aggregation methods that can handle arbitrary sizes. For example, an attention-based aggregation or similar is a parametric aggregation method but can handle arbitrary length inputs.

- The notation is a bit overwhelming and needs to be simplified and clarified. In Eq 5 alone, there are 4 different uses of the $\mathcal{L}$ with different subscripts.

- Choosing a mini-batch with the same missing features in a real-world VFL scenario would introduce bias in the training batch since it is unlikely for the missing features to be missing completely at random (MCAR). Rather, different groups of samples will likely have similar missing features. Can you explain how your method would handle this? This seems to be a severe limitation as the exact missingness pattern is unlikely to happen often unless the number of clients is very small.

- Related to above, it seems that this biasedness in mini-batches would cause Assumption 2 in your theory to be violated. Thus, it is unclear if your theory is reasonable for the practical algorithm and real-world situation.

- Relatedly, how do you handle the fact that there might be missingness patterns that are far more common than others?

- Does this assume that all clients participate during the training process (i.e., no client leaves even temporarily)? It seems that this requires all clients to be available during training (even if they do not participate in the training round).

- How are the baseline methods trained when there is missingness at test time? Are the batches with high missingness simply thrown out? It seems there are some simple baselines to adapt standard VFL by simply imputing zeros at train and test time or something like that. Stronger baselines with some naive adaptations like this for missing values would be helpful.  Currently, it seems that standard VFL is unfairly penalized.

- Do your experiments assume missingness completely at random (MCAR)? I'm guessing they do but wanted to double check. This would align with your theory.

- What happens if you do not weight things but just use simple dropout when sampling your missing features?

---

> ### Author Response · Authors · 2024-11-22
> **Official Comment by Authors (1/3)**
>
> Thank you for your comments.
>
> ### Re: combinatorial approach for a small number of clients
>
> * We have added new experiments comparing the combinatorial approach to our method and to the other baselines for a range of numbers of clients (see Section 5 of the revised manuscript). These experiments show that, even for a small number of clients, the combinatorial approach is significantly slower than our method, as seen in the following table, where we present the runtime of the combinatorial approach divided by the runtime of our LASER-VFL method for CIFAR-10:
>
>     | Number of Clients (K) | 2    | 3    | 4    | 5     | 6     | 7     |
>     |------------------------|------|------|------|-------|-------|-------|
>     | Runtime Ratio          | 1.76 | 3.19 | 5.96 | 10.91 | 19.41 | 35.80 |
>
>     This demonstrates that the combinatorial approach may become impractical even for a modest number of clients. Further, as mentioned below in our response, the combinatorial approach is in fact outperformed by our LASER-VFL method.
>
> ### Re: MCAR
>
> Thank you for raising this point. We have made it clearer in the revised manuscript that we do assume that the blocks of features are Missing Completely at Random (MCAR). As pointed out in a recent review paper on handling missing data [1] (in general machine learning; not specific to federated learning), the MCAR assumption is the most common type of assumption when dealing with missing data (compared to Missing At Random (MAR) and Missing Not At Random (MNAR)). This being said, it would be interesting to extend our convergence guarantees to the more challenging MAR setting, which is often more realistic.
>
> Our experiments in the original submission also assume MCAR and similar probabilities of missing for the different blocks of features. However, to address your concern, we are currently working on experiments with different missingness patterns to include in the revised manuscript.
>
> ### Re: VFL and GNNs
>
> Regarding your comments about the connection to GNNs and the use of parameter sharing, while we are not experts in GNNs (so please correct us if we interpreted your comment incorrectly), it seems to us that you are saying that we can interpret each client as a node in a complete graph and interpret the whole model as a GNN. If this is the case, we believe it may be possible to see our model as a single-layer GNN with average aggregation at each client’s own fusion predictor, yes. If our understanding of your comments is correct, a small correction on your comment about “sharing model parameters across nodes” would be that we do not share model parameters across nodes, but rather across tasks. Nevertheless, we do not claim that the idea of sharing parameters is novel. We simply claim that, by sharing model parameters across tasks and sampling a subset of tasks from the full set at each iteration, we obtain a good, novel solution to the problem of missing features in VFL. This solution is indeed simple, but we see that as a strength, not a weakness.
>
> ### Re: missing reference
>
> Thank you for bringing the work by Ganguli et al [2] to our attention, as indeed it is worth including in our paper as a related work. This being said, while [2] focuses on dealing with communication failure during inference for a network of edge devices in the cross-device FL (settings with a large number of clients), our work focuses on missing features during both training and inference.
>
> Note however that, while task sampling and dropout are related, they are not exactly the same. Our task-sampling mechanism allows the loss computed at each client to take into account multiple sets of tasks at each iteration (each task corresponding to prediction from a different subset of blocks of features), whereas dropout considers a single combination at each iteration. Further, our task-sampling mechanism takes into account the presence of missing training data (which is not considered in [2]); while in [2] the authors "simulate a dynamic graph using dropout", this is not equivalent to having missing blocks of features during training, as a missing block is missing _throughout_ training, rather than simply being absent from a given (set of) iteration(s).
>
> ### References
>
> [1] Zhou, Youran, Sunil Aryal, and Mohamed Reda Bouadjenek. "Review for Handling Missing Data with special missing mechanism." arXiv preprint arXiv:2404.04905 (2024).
>
> [2] Ganguli, Surojit, et al. "Fault-tolerant vertical federated learning on dynamic networks." arXiv preprint arXiv:2312.16638 (2023).

---

> ### Author Response · Authors · 2024-11-22
> **Official Comment by Authors (2/3)**
>
> ### Re: comparison with the combinatorial approach
>
> Regarding your comment that "it is difficult to argue that these models would be anywhere near the naive solution of actually directly training an exponential number of models", as mentioned above, we have added new experimental results in which we compare the combinatorial approach to our method and to the other baselines (see Section 5 of the revised manuscript).
>
> These new results show that our method in fact _outperforms_ the combinatorial approach. We attributed this to (1) the presence of missing features during training, meaning that in the combinatorial approach, for each batch, we can only train the predictors corresponding to a set of features that is a subset of the set of observed features, significantly decreasing the number of training samples and thus achieving worse generalization; and (2) even in the absence of missing training data, LASER-VFL still outperforms the combinatorial approach due to the dropout-like regularization effect introduced by task sampling.
>
> ### Re: notation complexity
>
> We have worked to simplify the notation in the revised manuscript (see the "Our optimization method" paragraph in Section 3.2). However, given the topic of the paper, further simplification is challenging, as the use of indices for sets of models, feature blocks, and stochastic approximations is often essential for clarity and precision.
>
> ### Re: client participation
>
> In this work, we did not consider communication faults nor clients leaving during training. In contrast to [2], which focuses on dynamic networks in cross-device VFL, our work focuses on the cross-silo setting, which is the most common setting in VFL [3], and tackle the fundamentally different problem of missing features in VFL, where the missing blocks of features are entirely absent from training, at all iterations.
>
> In cross-silo settings, there are fewer clients than in cross-device settings, typically with more reliable communication links. For example, as mentioned in the paper, an online retail company and a social media platform may hold different features on shared users. In such applications, communication failures are less problematic than in cross-device settings.
>
> Regarding your statement that, in our LASER-VFL method, all clients must be available during training, even if they do not participate in a round, we believe this is not the case. A client that does not observe its feature block in a batch does not need to join that communication round.
>
> ### Re: dealing with missing features in the standard VFL baseline
>
> The baseline standard VFL method does waste samples with missing features, making a random prediction instead. However, we did include other VFL-based baselines in the original submission, such as the ensemble method, which is still outperformed by our method, as is our newly added baseline: the combinatorial approach. While this is a strong baseline, it still performs worse than our LASER-VFL method, as mentioned above.
>
> ### Re: further baselines
>
> We have added a new baseline to the experiments in Section 5: PlugVFL [4]. PlugVFL is the most highly cited of the three works enabling inference in VFL from varying sets of feature blocks. After recently obtaining the code from the authors, we were able to include PlugVFL as a baseline in our experiments. While PlugVFL outperforms most baselines, it does so inconsistently and our LASER-VFL method ultimately outperforms PlugVFL, as shown in the revised Table 1 and in the experiments with a varying number of clients.
>
> Note that we have made the necessary extensions to PlugVFL to support missing training data and more than two clients, which were not considered in the approach described in the original paper. See Appendix D of the revised manuscript for details.
>
> We would also like to explain that the reason for not including more baselines in our original submission was the difficulty of reproducing results from the related methods, due to the unavailability of their source code. In this spirit, we are happy to share our code and make it open-source upon acceptance, to enable others to compare their methods with ours. In the meantime, we have made our code publicly available in an anonymos GitHub repository:
> https://anonymous.4open.science/r/robust-vfl-7A1E/README.md
>
> ### References
>
> [3] Liu, Yang, et al. "Vertical federated learning: Concepts, advances, and challenges." IEEE Transactions on Knowledge and Data Engineering (2024).
>
> [4] Jingwei Sun, Zhixu Du, Anna Dai, Saleh Baghersalimi, Alireza Amirshahi, Qilin Zheng, David Atienza, & Yiran Chen. (2024). PlugVFL: Robust and IP-Protecting Vertical Federated Learning against Unexpected Quitting of Parties.

---

> ### Author Response · Authors · 2024-11-26
> **Official Comment by Authors (3/3)**
>
> We have spent additional time understanding the paper by Ganguli et al. [2] and running new experiments. As a result, we have some further responses to share below.
>
> ### Re: comparison with Ganguli et al. [2]
>
> We would like to highlight that, to the best of our understanding, for the special case where all clients can always communicate with each other (a static, complete graph, which is the setup considered in our work), the method proposed by Ganguli et al. [2] matches the PlugVFL method proposed by Sun et al. [4], in the absence of the privacy mechanism therein. While [2] addresses the challenge of extending [4] to dynamic networks in the cross-device setting and thus has a different focus from that of our paper, our LASER-VFL method outperforms PlugVFL [4] in the special case of static networks in the cross-silo setting and, consequently, also outperforms [2] in this special case.
>
> ### Re: experiments with nonuniform missing features
>
> To address your question regarding how our method handles cases where some feature missing patterns are more common than others, we have added an experiment to the revised manuscript. See the "Nonuniform missing features" paragraph in Section 5 and Table 3 in Appendix E. In this experiment, we assume that each block $k$ has a probability of not being observed, $p_k$. Instead of setting all $\{p_k\}$ to the same value as in the previous experiments, we sample each $p_k$ independently from a beta distribution: $p_k \sim \text{Beta}(2.0, 2.0)$. As an example, for one of the seeds of our experiments with $K=4$ clients, we have $(p_1, p_2, p_3, p_4)=(0.43, 0.24, 0.46, 0.74)$. Our LASER-VFL method demonstrates consistent performance without any degradation and, as in the previous experiments, outperforms the baselines.
>
> ### References
>
> [2] Ganguli, Surojit, et al. "Fault-tolerant vertical federated learning on dynamic networks." arXiv preprint arXiv:2312.16638 (2023).
>
> [4] Jingwei Sun, Zhixu Du, Anna Dai, Saleh Baghersalimi, Alireza Amirshahi, Qilin Zheng, David Atienza, & Yiran Chen. (2024). PlugVFL: Robust and IP-Protecting Vertical Federated Learning against Unexpected Quitting of Parties.

---

> > ### Comment · Reviewer_wVzL · 2024-12-02
> >
> > Thank you for your work on the author response. The additional experiments and explanations are helpful so I'm inclined to increase my score by 1.
> >
> > I would note that I do not think PlugVFL is the same as [2] (even with static complete graph) because [2] also includes gossip ensembling across the clients' outputs, which is not done in PlugVFL and [2] shows that this aids in performance. That is at least one key difference between [2] and PlugVFL.
> >
> > Regarding GNNs/model sharing: My main concern was the novelty claim of this idea. Not that the idea itself is bad or overly simple. I just think claiming novelty on this point is an overemphasis given that model sharing is super common and well-known. I am happy that the idea is simple, but I think the novelty claim should be reduced significantly.

---

> > > ### Author Response · Authors · 2024-12-02
> > > **Thanks for the response**
> > >
> > > Thank you so much for your response and for raising the score; we truly appreciate it.
> > >
> > > Regarding the special case of static complete graphs, our understanding is that gossip, as defined in Ganguli et al. [2], leads to all clients obtaining the same aggregated vector through the gossiping process. We believe this results in identical outputs for gossip ensembling across the clients holding the aggregators.
> > >
> > > We have revised the paper to place less emphasis on the parameter-sharing aspect of our approach and will double-check to ensure that this is fully reflected in the final version.

---

> > > > ### Comment · Reviewer_wVzL · 2024-12-02
> > > >
> > > > Hi authors, Thanks for your response. I think I understand now that you are assuming that communication is always possible, just  that features are missing at some clients whereas as [2] and PlugVFL assume that communication may not always be possible. For [2], with a complete graph, all clients receive the same non-missing messages and thus could be expected to compute similar outputs. The outputs may not be exactly the same if the parameters at each aggregator are initialized differently but it is reasonable to assume they are close. Now I understand why PlugVFL and [2] are similar in this very special case of a static complete graph. Thanks for the clarification.

---

> > > > > ### Author Response · Authors · 2024-12-02
> > > > >
> > > > > Thanks for your prompt response! You have understood it perfectly—this is precisely the point we were aiming to convey.

---

> ### Author Response · Authors · 2024-11-28
> **Discussion Reminder**
>
> Dear Reviewer wVzL,
>
> Thank you again for your valuable comments and feedback. We hope that our response has addressed your concerns and clarified your questions. As the discussion period comes to a close, please feel free to reach out if you have any additional questions.
>
> Best regards,
> The Authors

---

> ### Author Response · Authors · 2024-12-01
> **Discussion Reminder**
>
> Dear Reviewer wVzL,
>
> Thank you again for your time and effort in reviewing our paper. Since tomorrow is the final day for reviewers to post comments, we would appreciate your feedback on whether our responses to your comments and revisions to the manuscript are satisfactory. If you have any further questions, please let us know and we will be happy to provide additional clarifications. If not, we would be grateful if you could consider increasing your score to reflect the fact that we have addressed your concerns.
>
> Best regards,
> The Authors

---

### Official Review · Reviewer_fR4j · 2024-11-04

**Soundness:** 3
**Presentation:** 2
**Contribution:** 2
**Rating:** 6
**Confidence:** 3

**Summary:**

This paper focuses on vertical federated learning (VFL), where data is partitioned by features rather than by samples. In this setting, each client possesses only a subset of the features for a specific sample. The paper addresses the challenge of missing features in the dataset and the heterogeneous availability of clients during the federation. Standard VFL approaches cannot handle these challenges, as it assumes that all features are available during both training and inference. To tackle these issues, this work proposes LASER-VFL, which relies on strategic sharing of model parameters and task sampling to train a family of predictors, allowing training and inference using any and all available blocks of features, leading to full utilization of the data. The paper provides theoretical guarantees for non-convex objectives and demonstrates empirical results on four different datasets, showing good performance.

**Strengths:**

The authors propose an approach for VFL that enables both training and inference using all available blocks of features.

The authors provide theoretical guarantees for convergence under L-smoothness, unbiasedness, and bounded variance assumptions.

The authors conduct empirical evaluations of the proposed methods over 4 datasets, showing good performance.

**Weaknesses:**

For VFL models, the authors mention that split neural networks are often considered and explains that they seek to find multiple parameters for each block and for a fusion model. Then mentions that they aim to solve (1) which requires all the blocks, which contradicts the paper objective seeking VFL with missing blocks. The authors should be clear from the early stage whether they aim to solve (1), or other works aimed to solve (1).

The authors mentions that LASER-VFL enables both training and inference
using all available blocks of features. However, in Figure 1 (b) (our method), shows that samples 4, 6, and 7, which has missing blocks, are only used for training (and not for testing), which is not clear why.

The authors go through some previous works very briefly in the introduction without going into details of their approaches and mention that these previous works add training stages and modules, making them more complex. While related work was provided later this paragraph seemed both redundant and unclear.

The authors mentioned some works that dealt with missing features. Then surprisingly claim that to the best of their knowledge, LASER-VFL is the first method to enables both training and inference using any and all available blocks of features. (line 108: We propose LASER-VFL (Leveraging All Samples Efficiently for Real-world VFL), a novel method that enables both training and inference using any and all available blocks of features. To the best of our knowledge, this is the first method to achieve this.)

Citations should be \citep when discussing some paper (not the authors). For example in line 144 it should be in (Feng & Yu 2020; Kang et al. 2022a) not in Feng & Yu (2020); Kang et al. (2022a).

In VFL problems what cases would really requires more than a few clients to collaborate? In the empirical setting the authors considered 4 clients. If we have a few companies, why not simply do a combinatorial approach? For such few client cases, say 2 or 3, the authors could show the optimal performance of training the 2^k-1 models as an optimal target against the proposed solution.

The authors introduce a lot of notations, which should be minimized to the least possible to make it easier to follow. (While the authors clearly mentions that the notation in (3) differs slightly from (1) and (2), this still should be avoided as much as possible).

While the authors mention several methods that dealt with missing blocks, most of these approaches were not considered in the empirical evaluation. Being more complex and requiring multiple stages or modules should not be a reason to skip the comparison. It could be that these added stages or complexities lead to better accuracy and/or better efficiency in terms of rounds to accuracy.

Giving uniform weights in the aggregation of the feature representations of different blocks of features for the different clients (Eq. (4)) would alter the performance compared to other concatenation methods. Averaging and sum can cancel some information which can be a drawback compared to concatenation methods.

While the authors claim that the method is suitable for a growing number of clients, avoiding the exponential growth in computation and memory. Empirical results should be provided to deal with more than 4 clients to see that the method indeed can achieve good performance. For example, would be nice to show for 8 and/or 16 clients.

The theory shows convergence in the non-convex setting, under L-smoothness and other assumptions, which is appreciated. However, such results do not reveal the optimality of the method.

**Questions:**

How would the performance of the aggregation mechanism differ when using weighted averaging rather than simple averaging? What if some clients have more powerful feature blocks? Giving uniform weights would alter the performance compared to other concatenation methods.

How would the performance be with even larger number of clients?

For small number of clients, the combinatorial approach becomes feasible, how does the combinatorial approach (which can be considered as an upper bound) differ from the proposed method?

How does the method perform against the other discussed related works which does solve the same problem but may have different training stages/modules?

How would the results be in a convex setting? Where does it converge? Does it guarantee optimal performance of VFL with missing clients?

---

> ### Author Response · Authors · 2024-11-22
> **Official Comment by Authors (1/2)**
>
> Thank you for your comments.
>
> ### Re: clarity when introducing Problem (1)
>
> We have made the sentence leading up to Problem (1) clearer.
>
> ### Re: missing feature blocks in Figure 1 (b)
>
> Samples 4, 6, and 7 in Figure 1(b) (our method) are not used by client 1 for inference because they correspond to samples where client 1 does not observe its own feature block. In the example in the paper of an online retailer and a social media company sharing users, this corresponds to the case where an online retailer does not perform the prediction task for an individual who is not one of its users.
>
> ### Re: mentioning prior art before the "Related work" paragraph
>
> The purpose of the "Dealing with missing features" paragraph in the introduction, which, as you noted, discusses previous works only at a high level, is to highlight that while there have been attempts to address this problem in VFL, none have fully solved it. This highlights the interest of the community on the topic and, more importantly, allows us to identify an important gap in the existing literature. In contrast, the related works section provides a more comprehensive review of the literature, including more detail on the approaches that these works have employed to address the issue of missing features in VFL.
>
> ### Re: claim of novelty
>
> We reiterate that, to the best of our knowledge, our LASER-VFL method _is_ the first method to enable both training _and_ inference using any and all available blocks of features. In contrast, the other works mentioned in the paper address either missing features during training _or_ missing features during inference, but not both. Our approach addresses both challenges simultaneously.
>
> ### Re: citations formatting
>
> The following quote is from the "Formatting Instructions for ICLR 2025 Conference Submissions": "When the authors _or the publication_ are included in the sentence, the citation should not be in parenthesis" (emphasis added by us).
>
> ### Re: combinatorial approach for a small number of clients
>
> We have added new numerical experiments (see Section 5 of the revised manuscript) comparing the combinatorial approach with our method and the other baselines. These experiments show that, even with a small number of clients, the combinatorial approach is significantly slower than our method, as seen in the following table, where we present the runtime of the combinatorial approach divided by the runtime of LASER-VFL for CIFAR-10:
>
> | Number of Clients (K) | 2    | 3    | 4    | 5     | 6     | 7     |
> |------------------------|------|------|------|-------|-------|-------|
> | Runtime Ratio          | 1.76 | 3.19 | 5.96 | 10.91 | 19.41 | 35.80 |
>
> Moreover, the combinatorial approach is not optimal in terms of performance: (1) in the presence of missing features during training, for each batch, it can only train the predictors using a set of features that is a subset of the set of observed features, significantly decreasing the number of training samples and thus achieving worse generalization; and (2) even when there is no missing training data, LASER-VFL still outperforms the combinatorial approach due to the dropout-like regularization effect introduced by task sampling.
>
> ### Re: notation complexity
>
> We have worked to simplify the notation in the revised manuscript. However, given the topic of the paper, further simplification is challenging, as the use of indices for sets of models, feature blocks, and stochastic approximations is often essential for clarity and precision. The choice to use slightly different notation in (3), compared to (1) and (2), was made to keep (1) and (2) as simple as possible. This approach helps ease the reader into the new notation before introducing the set notation, which is only necessary in (3). While consistency is important, we viewed this as a trade-off and prioritized simplicity where feasible.

---

> > ### Author Response · Authors · 2024-11-22
> > **Official Comment by Authors (2/2)**
> >
> > ### Re: baselines
> >
> > We compare our method to vanilla VFL and to a local approach, both of which are standard baseline methods to compare against [1, 2]. We also compare our method to VFedTrans for 2 datasets, despite it being more complex and requiring multiple stages, but observe that it performs only slightly better than the local approach and much worse than other baselines, such as ensemble. This motivated our focus on other baselines instead of VFedTrans. We have now also added a new baseline, the combinatorial approach, which one might even expect to act as an upper bound on performance. However, we observe in our experiments that our LASER-VFL method also beats the combinatorial approach, due to the aforementioned reasons.
> >
> > ### Re: on the choice of aggregation mechanism
> >
> > Your statement that addition and averaging lose information relative to concatenation is not very accurate. If the architecture of the representation model is designed together with the choice of aggregation mechanism, we can adjust the representation model so that using averaging instead of concatenation as the aggregation mechanism does not reduce the flexibility of the overall model, but simply shifts it from the fusion model to the representation models. We have made this clearer in the revised version of the manuscript (see Section 3.1). Allow us to elaborate on this.
> >
> > Consider the case where we have two clients, which extract representations $\mathbf{v}_1\in\mathbb{R}^{d_1}$ and $\mathbf{v}_2\in\mathbb{R}^{d_2}$. Denoting the first layer of the fusion model by $\mathbf{W}\in\mathbb{R}^{d_0\times(d_1+d_2)}$, with $\mathbf{W}=[\mathbf{W}_1 \text{ } \mathbf{W}_2]$, and defining $\mathbf{v}:=[\mathbf{v}_1^\top \text{ } \mathbf{v}_2^\top]^\top$, aggregation by concatenation allows for $\mathbf{W}\mathbf{v}=\mathbf{W}_1\mathbf{v}_1+\mathbf{W}_2\mathbf{v}_2$, whereas aggregation by sum restricts us to $\mathbf{W}_1(\mathbf{v}_1+\mathbf{v}_2)$. However, if we adjust the representation models 1 and 2 to include $\mathbf{W}_1$ and $\mathbf{W}_2$ respectively as the final layer of each, they will now output $\mathbf{W}_1\mathbf{v}_1$ and $\mathbf{W}_2\mathbf{v}_2$ respectively, allowing us to recover $\mathbf{W}_1\mathbf{v}_1+\mathbf{W}_2\mathbf{v}_2$ at the server while using aggregation by sum. We can therefore achieve the same flexibility as when employing aggregation by concatenation.
> >
> > This also addresses your question about the use of weighted averaging in the aggregation mechanism. To understand why, note that, since our model can learn $\mathbf{W}_1\mathbf{v}_1+\mathbf{W}_2\mathbf{v}_2$, it can learn a weighted average, which corresponds to $\mathbf{W}_i=w_i\mathbf{I}$, for $i\in\{1,2\}$, where $w_i\in\mathbb{R}$ and $\mathbf{I}$ is the identity matrix.
> >
> > We are currently working on a simple diagram to add to the supplementary material illustrating this.
> >
> > ### Re: experiments with $K>4$
> >
> > We have added new numerical experiments comparing the performance of our method with that of the combinatorial approach and of the other baselines for different numbers of clients, $K\in\{2,\dots,8\}$. These experiments show the improved performance of our method _across_ these different settings.
> >
> > ### Re: convergence guarantees
> >
> > The derived upper bound in Theorem 1 is order optimal for nonconvex functions [3]. On top of this, we guarantee convergence to the optimal function value under the PL condition, which is a subclass of nonconvex functions which also includes strongly convex functions as a special case.
> >
> > ### References
> >
> > [1] Kang, Yan, Yang Liu, and Xinle Liang. "FedCVT: Semi-supervised vertical federated learning with cross-view training." ACM Transactions on Intelligent Systems and Technology (TIST) 13.4 (2022): 1-16.
> >
> > [2] Huang, Chung-ju, Leye Wang, and Xiao Han. "Vertical federated knowledge transfer via representation distillation for healthcare collaboration networks." Proceedings of the ACM Web Conference 2023. 2023.
> >
> > [3] Arjevani, Yossi, et al. "Lower bounds for non-convex stochastic optimization." Mathematical Programming 199.1 (2023): 165-214.

---

> > > ### Comment · Reviewer_fR4j · 2024-11-23
> > >
> > > Thank you for your reply.
> > >
> > > I understand the ICLR citation formatting guidelines. However, if parentheses are not used, the phrasing can become grammatically incorrect or misleading. For example, can we say, "algorithm in Liu et al. 2020"? Does "algorithm in authors" make sense? If you wish to refer to the authors in a sentence, it should be phrased as "the algorithm in the work of Liu et al. (2020)" or "the algorithm in the work by Liu et al. (2020)." You may want to review this point further.
> > >
> > > Regarding the baselines, the response provided is unsatisfactory. We appreciate the addition of the combinatorial baseline which is an interesting baseline for scenarios involving a smaller number of clients. However, as mentioned in our review, although the authors reference several methods that address missing blocks, most of these approaches were not included in the empirical evaluation. The fact that these methods are more complex or require multiple stages or modules should not be a justification for excluding them. Added stages or complexities might contribute to better accuracy and/or improved efficiency in terms of rounds to accuracy. Hence, an empirical comparison with those is strongly encouraged.
> > >
> > > We further encourage the authors to clarify their discussion of related works and contributions. Criticizing prior works for adding complexities or modules is ambiguous; it is generally acceptable to introduce additional modules or complexities as long as the method remains efficient and effective. Moreover, incorporating additional baselines from the related works in the empirical analysis will undoubtedly add value to the paper and better showcase the performance of your proposed approach.
> > >
> > > The authors have addressed some of the concerns raised, including the addition of the combinatorial baseline, the consideration of a larger number of clients (K = 8), and responses to some questions, which is appreciated. Consequently, we will raise the score.

---

> > > > ### Author Response · Authors · 2024-11-24
> > > >
> > > > Many thanks for your reply and further suggestions.
> > > >
> > > > ### Re: citation formatting
> > > >
> > > > We have addressed your comment on the citation formatting and updated the revised manuscript accordingly.
> > > >
> > > > ### Re: further baselines
> > > >
> > > > We have added a new baseline to the experiments in Section 5: PlugVFL [4]. PlugVFL is the most highly cited of the three works enabling inference in VFL from varying sets of feature blocks. After recently obtaining the code from the authors, we were able to include PlugVFL as a baseline in our experiments. While PlugVFL outperforms most baselines, it does so inconsistently and our LASER-VFL method ultimately outperforms PlugVFL, as shown in the revised Table 1 and in the experiments with a varying number of clients.
> > > >
> > > > Note that we have made the necessary extensions to PlugVFL to support missing training data and more than two clients, which were not considered in the approach described in the original paper. See Appendix D of the revised manuscript for details.
> > > >
> > > > We would also like to explain that the reason for not including more baselines in our original submission was the difficulty of reproducing results from the related methods, due to the unavailability of their source code. In this spirit, we are happy to share our code and make it open-source upon acceptance, to enable others to compare their methods with ours. In the meantime, we have made our code publicly available in an anonymos GitHub repository:
> > > > https://anonymous.4open.science/r/robust-vfl-7A1E/README.md
> > > >
> > > > ### Re: more detailed discussion on related works
> > > >
> > > > We have added a new appendix (Appendix A in the current version) to the paper, to provide a more detailed comparison of our LASER-VFL method with the closest related works. In this appendix, we provide more details on these methods and highlight the distinction between our approach and existing methods.
> > > >
> > > >
> > > > ### References
> > > >
> > > > [4] Jingwei Sun, Zhixu Du, Anna Dai, Saleh Baghersalimi, Alireza Amirshahi, Qilin Zheng, David Atienza, & Yiran Chen. (2024). PlugVFL: Robust and IP-Protecting Vertical Federated Learning against Unexpected Quitting of Parties.

---

> > > > > ### Comment · Reviewer_fR4j · 2024-11-27
> > > > >
> > > > > Thank you for your response! I will increase my score.

---

> > > > > > ### Author Response · Authors · 2024-11-27
> > > > > >
> > > > > > Thank you so much for raising your score and for engaging in the discussion!
> > > > > >
> > > > > > In the meantime, we also thought more about our experiments, so we would like to share a further comment and clarification on the completeness of the baselines included in the experiments.
> > > > > >
> > > > > > ### Re: completeness of baselines included in the experiments
> > > > > >
> > > > > > On top of comparing our method to the `Local` approach and to `Stardard VFL`, which are standard baselines for work in this area, we have included a reasonable and effective baseline consisting of an `Ensemble` method.
> > > > > >
> > > > > > We then move to works that specifically target inference in VFL in the presence of missing features. First, we include `VFedTrans` [2], which is a recent method from the line of work that leverages information from the entire federation to train a local predictor (i.e., a predictor that depends only on the local features of its client during inference). We observe that, as expected, while such methods can slightly outperform the `Local` approach, they waste a lot of information when more feature blocks are observed during inference. For this reason, we see that the `Ensemble` baseline outperforms this method, as it leverages information from all the clients during inference.
> > > > > >
> > > > > > Afterwards, we compare with methods allowing for a varying number of feature blocks during inference. We start by including the `Combinatorial` baseline, which explicitly trains a predictor for each possible combination of missing features. We observe that, in addition to having much better scalability, our `LASER-VFL` method outperforms `Combinatorial`. Then, we look at the existing methods supporting a varying number of available feature blocks in the literature. To the best of our knowledge, there are only three papers dealing with a varying number of features blocks at inference time in VFL [4, 5, 6]. We have added `PlugVFL` [4] as a baseline, and have observed that our `LASER-VFL` method outperforms it. We also note that the paper by Ganguli et al. [5] is equivalent to `PlugVFL` in our setting (i.e., the case where all clients can always communicate with each other, which is prevalent in VFL). Finally, as we now explain in our revised manuscript (in Section 1), the work in [6] only considers binary classification and, since their algorithm relies on results from [7] which explicitly requires the assumption of binary classification, it is not straightforward to extend their method to non-binary tasks.
> > > > > >
> > > > > > Therefore, we believe that the _current baselines included in our experiments in Section 5 should be considered as complete_. These results allow us to claim that our method outperforms state-of-the-art methods for related problems in the presence of missing features. This is a significant and timely contribution to VFL, which we believe makes our work worthy of a (more than borderline) acceptance.
> > > > > >
> > > > > > Thank you again and please let us know if you have any further comment or question.
> > > > > >
> > > > > > ### References
> > > > > >
> > > > > > [2] Huang, Chung-ju, Leye Wang, and Xiao Han. "Vertical federated knowledge transfer via representation distillation for healthcare collaboration networks." Proceedings of the ACM Web Conference 2023. 2023.
> > > > > >
> > > > > > [4] Jingwei Sun, Zhixu Du, Anna Dai, Saleh Baghersalimi, Alireza Amirshahi, Qilin Zheng, David Atienza, & Yiran Chen. (2024). PlugVFL: Robust and IP-Protecting Vertical Federated Learning against Unexpected Quitting of Parties.
> > > > > >
> > > > > > [5] Ganguli, Surojit, et al. "Fault-tolerant vertical federated learning on dynamic networks." arXiv preprint arXiv:2312.16638 (2023).
> > > > > >
> > > > > > [6] Gao, Dashan, et al. "Complementary Knowledge Distillation for Robust and Privacy-Preserving Model Serving in Vertical Federated Learning." Proceedings of the AAAI Conference on Artificial Intelligence. Vol. 38. No. 18. 2024.
> > > > > >
> > > > > > [7] Zhang, Harry. "The Optimality of Naive Bayes." Proceedings of the the 17th International FLAIRS conference (FLAIRS2004). 2004.

---

### Official Review · Reviewer_DiU7 · 2024-11-06

**Soundness:** 3
**Presentation:** 2
**Contribution:** 3
**Rating:** 6
**Confidence:** 2

**Summary:**

This paper studies vertical FL with missing features. The authors propose a new vertical FL method named **LASER-VFL** for efficient training and inference of split neural network-based models. This method relies on the strategic sharing of model parameters and on task-sampling to train a family of predictors. The authors also provide convergence rates for **LASER-VFL** under different objective functions.

**Strengths:**

1. This paper proposes a new algorithm to address the missing features in vertical FL.

2. Valid convergence analysis and experiments are provided.

**Weaknesses:**

1. The notations are too complex, which is not friendly for readers. The authors should try to simplify notations.

2. I'm not an expert in vertical FL, so I'm confused about the **key idea** presented on page 4.  So it will be better to add workflow diagrams to explain the method.

3. The experiments are conducted on the data with synthetic missing features. It would be more convincing to show results on real-world vertical FL data with missing features.

**Questions:**

1. What do hyperparameters refer to in vertical FL? Is it the optimization parameters such as learning rate or some specific parameter in vertical FL?

2. Does the proposed algorithm allow multiple local updates in each machine like traditional FL?

---

> ### Author Response · Authors · 2024-11-22
>
> Thank you for your comments.
>
> ### Re: notation complexity
>
> We have worked to simplify the notation in the revised manuscript (see the "Our optimization method" paragraph in Section 3.2). However, given the topic of the paper, further simplification is challenging, as the use of indices for sets of models, feature blocks, and stochastic approximations is often essential for clarity and precision.
>
> ### Re: workflow diagram
>
> Thank you for your suggestion of the workflow diagram. We have added a diagram illustrating our LASER-VFL method in Appendix B of the revised manuscript (as space constraints prevented its inclusion in the main paper) and referenced it just before introducing the **key idea** on page 4.
>
> ### Re: synthetic missing features
>
> We agree that running experiments on real vertical FL data with missing features would be ideal. However, such data is typically privately held by companies and is not publicly accessible. Consequently, as is standard practice in this area [1, 2, 3], we have used publicly available datasets, synthetically partitioned the data across clients, and simulated missing feature patterns.
>
> ### Re: hyperparameters in vertical FL
>
> If your question regarding hyperparameters refers to our statement in the contributions bullet points that our LASER-VFL method is "hyperparameter-free", we would like to clarify that we were not implying the removal of any hyperparameters specific to vertical FL. Rather, we meant that, unlike other works addressing missing features in vertical FL [1, 2, 3], LASER-VFL does not introduce any additional hyperparameters. Please let us know if we have misunderstood your question.
>
> ### Re: multiple local updates
>
> The proposed algorithm does not consider multiple local updates. Extending LASER-VFL to accommodate multiple local updates would be an interesting direction for future work. However, it is important to note that the use of multiple local updates in vertical FL, while possible [4], is not as prevalent as in horizontal FL. The main reason for this is that performing multiple local updates requires all clients to have access to the labels used for computing the loss, as well as the fusion model.
>
> To illustrate this, consider a simplified vertical FL model represented as a composition $g \circ f$, where $g$ is the fusion model and $f$ represents a local representation model. After the first local update of $f$, further local updates would require additional forward and backward passes over both $f$ and $g$ at the client, including the computation of the loss $\ell(y^n,g(f(x^n)))$. In contrast, by using a single update, LASER-VFL enables clients to use their (possibly different) labels locally and communicate only the derivative of the loss with respect to the features extracted by each representation model.
>
> ### References
>
> [1] Kang, Yan, Yang Liu, and Xinle Liang. "FedCVT: Semi-supervised vertical federated learning with cross-view training." ACM Transactions on Intelligent Systems and Technology (TIST) 13.4 (2022): 1-16.
>
> [2] He, Yuanqin, et al. "A hybrid self-supervised learning framework for vertical federated learning." IEEE Transactions on Big Data (2024).
>
> [3] Huang, Chung-ju, Leye Wang, and Xiao Han. "Vertical federated knowledge transfer via representation distillation for healthcare collaboration networks." Proceedings of the ACM Web Conference 2023. 2023.
>
> [4] Liu, Yang, et al. "FedBCD: A communication-efficient collaborative learning framework for distributed features." IEEE Transactions on Signal Processing 70 (2022): 4277-4290.

---

> > ### Comment · Reviewer_DiU7 · 2024-11-27
> >
> > Thank you for the response! I will keep my score.

---

### Meta-Review · Area_Chair_XWSc · 2024-12-20

**Metareview:**

The authors propose a new vertical federated learning (FL) method named LASER-VFL, designed for efficient training and inference of split neural network-based models in vertical FL settings with missing features. LASER-VFL addresses the challenge of missing labels by training a set of representation models (capable of performing well across different combinations of feature blocks) and a fusion model on each client (to handle any combination of representations).

The approach is interesting and compelling, and the numerical experiments verify its effectiveness. The paper is also complemented by a theoretical analysis under standard assumptions, such as smoothness and the PL condition. However, Theorems 1 and 2 provide standard convergence results for unbiased stochastic gradient descent, and their connection to the proposed algorithm remains underexplored. For instance, the presented results do not clarify how the number of clients $K$ affects the convergence guarantees, nor do they provide insights into the quality of the fusion or representation models. I believe the contribution would be significantly stronger if the theoretical analysis offered concrete guidelines and insights into the scalability of the method. Please add a discussion on this to the final version of the work.

Although the proposed method is not yet fully supported by the theory, it is intriguing and novel, and it has the potential to inspire further research within the vertical FL community. Thus I recommend acceptance.

**Additional Comments On Reviewer Discussion:**

The discussion between the authors and the reviewers addressed some concerns about the paper and led to the recommendation to include stronger baselines in the numerical comparison.

There was also an internal discussion regarding the complexity of the notation. Specifically, it was somewhat unclear how the Missing Completely at Random (MCAR) assumption was applied in the definition of the objective function and in the proof of Lemma 1. We encourage the authors to clarify the notation and provide additional explanations to improve the paper’s clarity.

---

### Decision · Program_Chairs · 2025-01-22

Accept (Poster)